# KEY-GRAPH TRANSFORMER FOR IMAGE RESTORATION

## ABSTRACT

It is widely acknowledged that capturing non-local information among pixels within one input image is crucial for effective image restoration (IR). However, fully incorporating such global cues into transformer-based methods can be computationally expensive, mainly when dealing with large input images or patches. Furthermore, it is assumed that the attention mechanism within the transformer considers numerous unnecessary global cues of the pixels from unrelated objects or regions. In response to these challenges, we introduce the Key-Graph Transformer (KGT) for IR in this paper. Specifically, KGT treats image features within a given window as the nodes of a graph. Instead of establishing connections among all the nodes, the proposed Key-Graph Constructor creates a sparse yet representative Key-Graph that flexibly connects only the essential nodes. Then the Key-Graph Attention Block is proposed within each KGT layer to conduct the self-attention operation guided by the Key-Graph only among selected nodes with linear computational complexity. Extensive experimental results validate that the proposed KGT outperforms state-of-the-art methods on various benchmark datasets, quantitatively and qualitatively.

## 1 INTRODUCTION

Image restoration (IR), a fundamental task in the realm of low-level computer vision, is dedicated to the improvement of images that have been compromised by various factors such as noise, blur, or other forms of distortion. The central aim of image restoration is to reconstruct a cleaner, visually more appealing version of the original image, thus facilitating a more effective analysis and interpretation. This capability finds diverse applications, including information recovery (such as retrieving obscured data in medical imaging, surveillance, and satellite imagery) and supporting downstream vision tasks like object detection, recognition, and tracking Sezan & Stark (1982); Molina et al. (2001). Despite significant advancements in recent years, it is noteworthy that current popular image restoration methods still face challenges in effectively handling complex distortions or preserving/recovering essential image details Li et al. (2023a). In order to recover high-quality images, the rich information exhibited in the degraded counterparts needs to be exquisitely explored.

For IR in modern computer vision systems, the de-facto representative networks are mainly built based on three fundamental architectural paradigms, *i.e.,* the convolutional neural networks (CNNs) LeCun et al. (1998); Zamir et al. (2021), Vision Transformers (ViTs) Vaswani et al. (2017); Dosovitskiy et al. (2020) and the Multilayer perceptrons (MLPs) Bishop & Nasrabadi (2006); Tu et al. (2022). The input image/image patches are treated as a regular grid of pixels in the Euclidean space for CNNs or a sequence of patches for ViTs and MLPs. However, the degraded input images usually contain irregular and complex objects. These architectural choices perform admirably in specific scenarios characterized by regular or well-organized object boundaries but have limitations when applied to images with more flexible and complex geometrical contexts.

Besides the above-mentioned limitations of how they treat data, CNNs are struggling to model the long-range dependencies because of their limited receptive field. Though ViTs have been validated as highly effective in capturing the long-range relation among pixels with the multi-head self-attention mechanism (MSA) Vaswani et al. (2017); Dosovitskiy et al. (2020); Ren et al. (2023a), their computational complexity increases quadratically with respect to spatial resolution. Similarly, MLPs are not trivial to be applied to high-resolution input spatial-wise, which largely reduces the

ability of MLPs to maintain the locality among the neighbor pixels. To overcome these limitations, recent methods investigate strategies for complexity reduction. One common approach is to implement MSA within local image regions using the SWIN-Transformer architecture design Liang et al. (2021); Li et al. (2023a). However, these designs treat input still as sequences, which hinders effective context aggregation within local neighborhoods and struggles to capture inherent connections among irregular objects. Additionally, an earlier study Zontak & Irani (2011) highlights that smooth image contents occur more frequently than complex image details, suggesting the need for differentiated treatment for different contents.

In this paper, we introduce a novel approach, the Key-Graph Transformer (KGT), to address the aforementioned limitations using the SWIN Liu et al. (2021) architecture. Our method comprises two core components: a K-nearest neighbors (KNN) based Key-Graph Constructor and a Key-Graph Transformer layer with a novel Key-Graph Attention block integrated. Specifically, starting with the input feature obtained from the convolutional feature extractor within one window, we treat each of them as a node representation of a graph. Since capturing long-range dependencies among all nodes can be highly computationally demanding, we selectively choose topK nodes based on the proposed Key-Graph constructor rather than establishing connections between all possible nodes. In particular, we propose a random topK strategy during training instead of a fixed topK value. This leads to a sparse yet representative graph that connects only the essential neighbor nodes, which makes our method achieve the same receptive field as previous transformer-based methods while effectively maintaining lower computational costs. The criteria for selecting these representative nodes are determined by the self-similarity calculated at the beginning of each KGT layer. Then the chosen nodes undergo processing by all the successive Key-Graph transformer layers shown in Fig. 1. It's worth noting that the implementation of the Key-Graph attention block within each KGT layer is achieved in three manners (*i.e.,* the Triton Dao et al. (2022), torch-mask, and torch-gather), which will be discussed in our ablation studies. Based on these two components, the information that exists in all the selected nodes is well-aggregated and updated.

In summary, our main contributions can be categorized as follows:

1. We propose a Key-Graph constructor that provides a sparse yet representative Key-Graph with the most relevant K nodes considered, which works as a reference for the subsequent attention layer, facilitating more efficient attention operations.

2. Based on the constructed Key-Graph, we introduce a Key-Graph Transformer layer with a novel Key-Graph attention block integrated. Notably, the computational complexity can be significantly reduced from quadratic to linear when compared to conventional attention operations.

3. Based on both the Key-Graph constructor and the Key-Graph Transformer layer, we propose a Key-Graph Transformer (KGT) for image restoration. Extensive experimental results show that the proposed KGT achieves state-of-the-art performance on 6 IR tasks.

## 2 RELATED WORK

**Image Restoration (IR),** as a long-standing ill-posed inverse problem, is designed to reconstruct the high-quality image from the corresponding degraded counterpart. It has been brought to various real-life scenarios due to its valuable application property Richardson (1972); Banham & Katsaggelos (1997); Li et al. (2023b). Initially, IR was addressed through model-based solutions, involving the search for solutions to specific formulations. However, with the remarkable advancements in deep neural networks, learning-based approaches have gained increasing popularity. These approaches have been explored from various angles, encompassing both regression-based Lim et al. (2017); Liang et al. (2021); Chen et al. (2021b); Li et al. (2023a) pipelines and generative model-based pipelines Gao et al. (2023); Wang et al. (2023); Luo et al. (2023); Yue et al. (2023). In this paper, our primary focus is to investigate IR within the context of the regression-based pipeline.

**Non-local Priors Modeling in IR.** Tradition model-based IR methods reconstruct the image by regularizing the results (*e.g.*, Tikhonov regularization Golub et al. (1999)) with formulaic prior knowledge of natural image distribution. However, it's challenging for these model-based methods to recover realistic detailed results with hand-designed priors. Besides, some other classic method finds that self-similarity is an effective prior which leads to an impressive performance Buades et al. (2005); Dabov et al. (2007). Apart from the traditional methods, the non-local prior also has been uti-

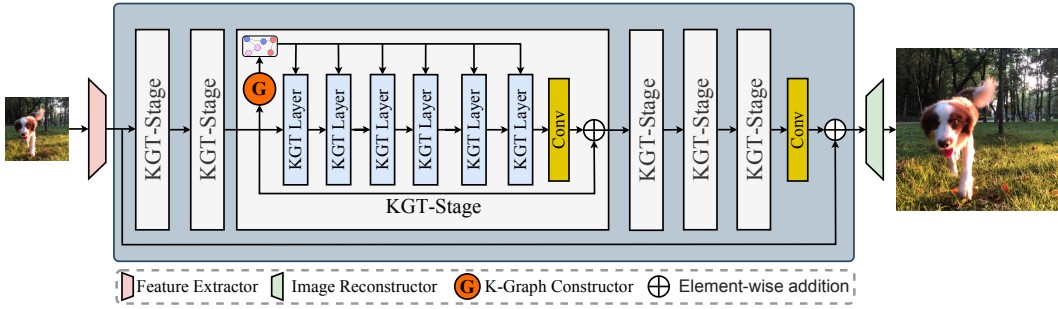

Figure 1: The architecture of the proposed Key-Graph Transformer (KGT) for Image Restoration. KGT mainly consists of a feature extractor, the main body of the proposed KGT (The main body here is for SR, while the U-shaped structure is used for other IR tasks), and an image reconstructor.

lized in modern deep learning networks Liu et al. (2018); Wang et al. (2018); Li et al. (2023a); Zhang et al. (2019b), and it was usually captured by the self-attention mechanism. Especially, KiT Lee et al. (2022) proposed to increase the non-local connectivity between patches of different positions via a KNN matching to better capture the non-local relations between the base patch and other patches in every attention operation, this brings more extra computation costs. DRSformer Chen et al. (2023) proposed a topK selection strategy that chooses the most relevant tokens to model the non-local priors for draining after each self-attention operation without reducing the computation complexity. The effectiveness of non-local priors has been widely validated in various recent transformer-based IR methods Liang et al. (2021); Zamir et al. (2022); Li et al. (2023a).

**Graph-Perspective Solutions for IR.** Graph operations are usually used to deal with irregular data structures such as point clouds Wang et al. (2019); Li et al. (2021b), social networks Myers et al. (2014), or protein desins Ingraham et al. (2019). Recently, graph-based methods were also adapted to process the input image/patches in a more flexible manner Gori et al. (2005); Scarselli et al. (2008); Mou et al. (2021); Han et al. (2022); Jiang et al. (2023) on various IR tasks, like facial expression restoration Liu et al. (2020), image denoising Simonovsky & Komodakis (2017), and artifact reduction Mou et al. (2021). However, most of the previous graph-based solutions for IR mainly extend from graph neural networks (GNNs), which mainly focus on very close neighbor nodes. Merely increasing the depth or width of GNNs proves inadequate for expanding receptive fields Xu et al. (2018), as larger GNNs often face optimization challenges like vanishing gradients and over-smoothing representation. Jiang et al. (2023) construct the graph with transformer-based architecture but in a very expensive manner where each node is connected to all other nodes. In this paper, we introduce a novel approach that integrates graph properties into ViTs by employing a Key-Graph for the efficient capture of effective non-local priors in Image Restoration (IR) tasks.

## 3 METHODOLOGY

The overall architecture of the proposed Key-Graph Transformer (KGT) is shown in Fig. 1. Unlike conventional approaches that treat input features after the convolutional feature extractor as a regular grid of pixels in Euclidean space (typical in CNNs) or as a sequence of patches (common in ViTs and MLPs), our method adopts a more flexible approach based on graph representation. To be specific, the proposed KGT focuses on enhancing the efficiency of representation learning in a multi-stage manner. The graph structure is shared within each KGT stage and can be dynamically updated at the beginning of each KGT stage, which leads to a sparse yet highly effective node representation. Before delving into our proposed method, we begin by offering a succinct overview of the foundational concepts of graph transformers in the preliminary section (Section 3.1). We then ensure the efficiency of graph updating by introducing the Key-Graph constructor (Section 3.2). Simultaneously, we attain the effectiveness of node feature aggregation by employing the Key-Graph Transformer Layer (Section 3.3) in each stage of the KGT.

### 3.1 PRELIMINARY: GRAPH TRANSFORMER.

In conventional vision transformers, graph nodes are typically assigned based on feature patches, and graph edges are usually represented by inferring similarities among nodes using a self-attention

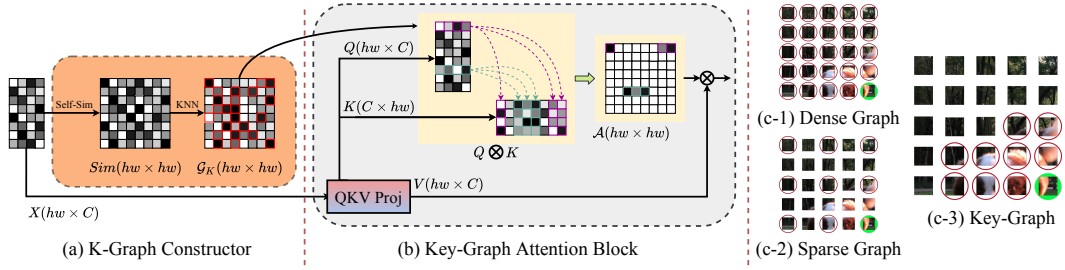

Figure 2: (a): The illustration of Key-Graph Constructor. (b): The proposed Key-Graph Attention Block within each KGT Layer. (c): The depiction of the difference among dense graph attention c-1, sparse graph attention c-2, and the proposed Key-Graph attention c-3.

mechanism Vaswani et al. (2017). Specifically, given the low-level input feature $F_{in} \in \mathbb{R}^{H \times W \times C}$, where $H$, $W$, and $C$ denote the height, the width, and the numbers of channels of the given feature, respectively. $F_{in}$ is split into $N$ feature patches and we get the node feature representation $X = \{x_i | x_i \in \mathbb{R}^{hw \times c}, i = 1, 2, 3, ..., N\}$, where $h$, $w$, and $c$ denote the height, the width, and the feature dimension of each patched feature. These features can be considered to be an unordered set of nodes. For each node $x_i$, an edge $e_{ji}$ can be added from $x_j$ to $x_i$ from all the neighbors of $x_i$ in $X$. Thus, a graph $\mathcal{G}$ is naturally constructed and can be represented by the corresponding adjacency matrix $\mathcal{A} = \{e_{ji}\} \in \mathbb{R}^{N \times N}$.

In order to get $\mathcal{A}$, we begin by linearly projecting $X$ into Query ($Q$), Key ($K$), and Value ($V$) matrices (note that $V$ will be used to conduct the node feature aggregation with the help of $\mathcal{A}$ later), which are denoted as $Q = X\mathbf{W}_Q$, $K = X\mathbf{W}_K$, and $V = X\mathbf{W}_V$, respectively. Here, $\mathbf{W}_{Q/K/V}$ represents the learnable projection weights. The calculation of $\mathcal{A}$ is performed as follows:

$$\mathcal{A}_{ij} = softmax(QK^T/\sqrt{d}) = \frac{exp(Q_i K_j/\sqrt{d})}{\sum_{k \in X_i} exp(Q_i K_k/\sqrt{d})}, j \in X_i \tag{1}$$

where $d$ represents the dimension of Q/K. Then the node feature can be aggregated to $\hat{x}_i$ by:

$$\hat{x}_i = \mathcal{A}_{ij} \cdot V_i \tag{2}$$

Since we have adopted the SWIN transformer Liu et al. (2021) as the basic architecture that conducts the window-wise attention, all the operations within each window are similar. To streamline our explanation, we select a single window for illustration when discussing the Key-Graph Constructor and the proposed Key-Graph Transformer layer. Notably, notations such as $F_{in}$ and $X$ are all window-size adapted for clarity.

## 3.2 KEY-GRAPH CONSTRUCTOR

The goal of the proposed Key-Graph constructor is to construct a sparse yet representative graph $\mathcal{G}_K$ at the beginning of each KGT stage. Specifically, given node representation $X$ of $F_{in}$, an initial fully connected Graph $\mathcal{G}$ is constructed by calculating the self-similarity of $X$ via dot product operation. As a result, the corresponding adjacency metrics $\mathcal{A}$ can be achieved:

$$\mathcal{A} = sim(i, j) = x_i \cdot x_j^T, \tag{3}$$

which describes the correlation among all the nodes, and a higher similarity value indicates a higher correlation between two nodes. However, in this context, $\mathcal{A}$ represents a fully connected graph, wherein all nodes $x_j$ within $X$ are included in the connectivity of the destination node $x_i$, irrespective of the degree of relatedness between $x_i$ and $x_j$ (e.g., Fig. 2 (c-1) describes such a case that the green dog patch node at the bottom right corner is also connected to all other nodes with a red circle marked. Best viewed by zooming.).

To mitigate the side effects of nodes with low correlation (e.g., the tree-related nodes at the upper left part) for the destination dog node, we propose to keep only K highly related nodes of the destination node $x_i$ and exclude the remaining nodes with low correlation. This is achieved under the guidance

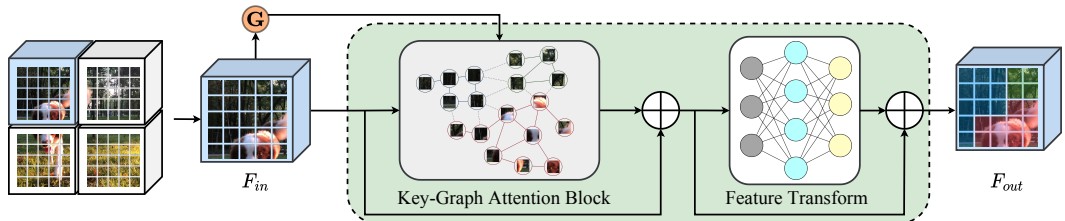

Figure 3: The framework of the proposed Key-Graph Transformer Layer.

of the similarity value from $\mathcal{A}$ as follows:

$$\mathcal{A}_K(i,j) = \begin{cases} \mathcal{A}(i,j), & \mathcal{A}(i,j) \geq sim(i,)_K \\ 0, & else \end{cases} \tag{4}$$

where $sim(i,)_K$ denotes the $K_{th}$ largest connective value of node $x_i$ with its corresponding node. As a result, once $\mathcal{A}_K$ is achieved, we get a sparse yet representative Key-Graph $\mathcal{G}_K$ which contains only the edge connection among the destination node (*e.g.*, the green dog node) and the other nodes with high correlation (*e.g.*, dog-related nodes. An example is illustrated in Fig. 2 (c-3)).

Owing to the permutation-invariant property inherent in both the MSA and the FFN within each transformer layer Vaswani et al. (2017); Lee et al. (2019), the KGT layer consistently produces identical representations for nodes that share the same attributes, regardless of their positions or the surrounding structures within the graph Chen et al. (2022a). In other words, nodes at the same spatial location are consistently connected to other nodes possessing the same attributes as they traverse through the various layers within the same KGT stage. This enables our Key-Graph $\mathcal{G}_K$ to act as a reference for each attention block in the subsequent KGT layers within each KGT stage, facilitating more efficient attention operations. This is different from the sparse graph (illustrated in Fig. 2 (c-2)) that only activates the nodes in a fixed coordinate of a given feature map Zhang et al. (2023b).

## 3.3 KEY-GRAPH TRANSFORMER LAYER

The proposed Key-Graph Transformer Layer is shown in Fig. 3, which consists of a Key-Graph attention block together with an FFN for the node feature aggregation. Fig. 2 (b) shows the detailed workflow of the proposed Key-Graph attention block. Initially, the node features $X$ undergo a linear projection into $Q$, $K$, and $V$. Then for each node in $x_i$ in $Q$, instead of calculating the self-attention with all the nodes $x_j$ in $K$, we choose only topK nodes in $K$ where $j$ denotes only the index of the nodes with high correlation to the given destination node. The selection is guided by the Key-Graph $\mathcal{G}_K$. We intuitively show such a process in Fig. 2 (a) and (b), and formulate the selection process as $K_K = select(topK, \mathcal{G}_K)$. Then the spare yet representative adjacency matrix $\mathcal{A}_K$ is can be obtained by:

$$\mathcal{A}_K = softmax_K(QK_K^T/\sqrt{d}), \tag{5}$$

which captures the pair-wise relation between each destination node $x_i, (i = 1, 2, ..., hw)$ in $Q$ with only the nodes that are semantically related to the given $x_i$. For other nodes apart from the selected K nodes, we keep their position in their corresponding places without any computation. This is different from the conventional self-attention operation which calculates the relation of each node in $Q$ and all nodes in $K$ (The difference between c-1 and c-3 in Fig. 2). Meanwhile, the proposed method is also different from the sparse attention used in Zhang et al. (2023b) where the position of the nodes that need to be collected is always fixed (The difference between c-2 and c-3 in Fig. 2). Conversely, the proposed Key-Graph attention block not only significantly reduces the computational complexity from $\mathcal{O}((hw)^2)$ to $\mathcal{O}((hw) \times K)$, where $K < hw$, but also provides a more flexible approach to capturing semantically highly related nodes.

Note that since the dimension of the select $\mathcal{G}_K$ only contains topK nodes, this leads to a dimension mismatch problem for the conventional self-attention mechanism. As a remedy, we tried three different manners for the detailed implementation, *i.e.*, (i) *Triton*, (ii) *Torch-Gather*, and (iii) *Torch-Mask*. Specifically, (i) is based on FlashAttention Dao et al. (2022), and parallel GPU kernels are called for the nodes. (ii) means that we use the 'torch.gather()' function in PyTorch to choose the

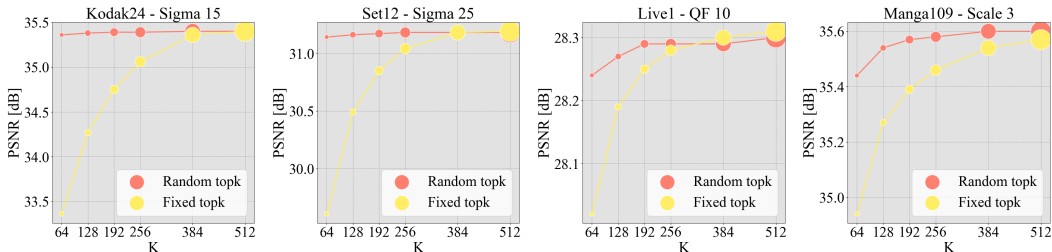

(a) Color image denoising on Kodak24, $\sigma = 15$.  (b) Grayscale image denoising on Set12, $\sigma = 25$  (c) Color image JPEG CAR on Live1, $QF = 10$.  (d) SR with KGT-B on Manga109, upscaler $\times 3$.

Figure 4: Ablation study on the impact of $K$. The size of the circle denotes the FLOPs.

corresponding $Q_{gather}$ and $K_{gather}$ based on $\mathcal{G}_K$, then the attention operation shown in Eq. 5 is conducted between $Q_{gather}$ and $K_{gather}$. (iii) denotes that we keep only the value of selected nodes of $\mathcal{A}_K$ and omitting other nodes with low correlation via assigning those values to $-\infty$ guided by $\mathcal{G}_K$. We will discuss the pros and cons of these manners in Sec. 4.1.

Finally, as the phenomenon of over-smoothing is prevalent in graph-structured data, it becomes particularly pronounced in deep models Chen et al. (2020); Keriven (2022). To relieve the over-smoothing phenomenon and encourage the node feature transformation capacity, we aggregate the node feature by an FFN in a residual connection manner. This process can be formalized as follows:

$$Z = FFN(X + \mathcal{A}_K V), \tag{6}$$

where $Z = \{z_i | z_i \in \mathbb{R}^{hw \times c}, i = 1, 2, 3, ..., N\}$ is the aggregated node feature.

## 4 EXPERIMENTS

In this section, we first analyze two important aspects of the proposed KGT, followed by extensive experiments on **6** IR tasks, which include JPEG compression artifact reduction (CAR), image denoising, demosaicking, IR in adverse weather conditions (AWC), image super-resolution (SR), and image deblurring. More details about the training protocols, the training/testing dataset, and additional visual results are shown in the *supplementary material (Supp. Mat.)*. In addition, the best and the second-best quantitative results in all tables are reported in red and blue, respectively.

### 4.1 ABLATION STUDY

Extensive ablation experiments are conducted for the following two essential explorations:

**The Impact of the K in Key-Graph Constructor.** Two sets of experiments are conducted to study the influence of hyper-parameter K. In the first set, K was held constant at 512 throughout the training process, while in the second set, K was randomly sampled from the values $[64, 128, 192, 256, 384, 512]$. It's important to note that for both sets, K was configured to the specified value during the inference phase. Besides the truth that the computational complexity grows linearly with K, there are two interesting phenomena that can be observed from the results shown in Fig. 4, *i.e.,* (1). The randomly sampled strategy has a very stable and better performance compared to the fixed K manner especially when the K is fixed to a small number (*i.e.*, 64, 128, 256). (2) The PSNR can largely increase with the increase of K in a fixed manner. We conclude that a random sampled strategy is more general and stable. It can also make the inference process more flexible regarding different computation resources. More ablation results can be found in our *Supp. Mat.* about the effect of the noise level and quality factor for denoising and JPEG CAR.

**The impact of the implementation of Key-Graph Attention** is assessed in terms of (i) *Triton*, (ii) *Torch-Gather*, and (iii) *Torch-Mask* under different numbers of N (various from 512 to 8192) and K (various from 32 to 512). The results of the GPU memory footprint are shown in Tab. 3, which indicate that *Torch-Gather* brings no redundant computation while requiring a large memory footprint. Though *Torch-Mask* brings the GPU memory increase, the increment is affordable compared to *Torch-Gather* and also very easy to implement. *Triton* can largely save the GPU memory while at the cost of slow inference and difficult implementation for the back-propagation process. To

Table 1: *Grayscale image JPEG compression artifact removal* results. †A single model is trained to handle multiple quality factors.

| Set | QF | JPEG PSNR | JPEG SSIM | †DnCNN3 PSNR | †DnCNN3 SSIM | †DRUNet PSNR | †DRUNet SSIM | †KGT (Ours) PSNR | †KGT (Ours) SSIM | GRL-S PSNR | GRL-S SSIM | SwinIR PSNR | SwinIR SSIM | ART PSNR | ART SSIM | CAT PSNR | CAT SSIM | KGT (Ours) PSNR | KGT (Ours) SSIM |
|---|---|---|---|---|---|---|---|---|---|---|---|---|---|---|---|---|---|---|---|
| Classic5 | 10 | 27.82 | 0.7600 | 29.40 | 0.8030 | 30.16 | 0.8234 | 30.26 | 0.8240 | 30.20 | 0.8286 | 30.27 | 0.8249 | 30.27 | 0.8258 | 30.26 | 0.8250 | 30.36 | 0.8267 |
| | 20 | 30.12 | 0.8340 | 31.63 | 0.8610 | 32.39 | 0.8734 | 32.52 | 0.8740 | 32.49 | 0.8776 | 32.52 | 0.8748 | - | - | 32.57 | 0.8754 | 32.58 | 0.8748 |
| | 30 | 31.48 | 0.8670 | 32.91 | 0.8860 | 33.59 | 0.8949 | 33.74 | 0.8955 | 33.72 | 0.8985 | 33.73 | 0.8961 | 33.74 | 0.8964 | 33.77 | 0.8964 | 33.77 | 0.8958 |
| | 40 | 32.43 | 0.8850 | 33.77 | 0.9000 | 34.41 | 0.9075 | 34.55 | 0.9078 | 34.53 | 0.9107 | 34.52 | 0.9082 | 34.55 | 0.9086 | 34.58 | 0.9087 | 34.57 | 0.9080 |
| LIVE1 | 10 | 27.77 | 0.7730 | 29.19 | 0.8120 | 29.79 | 0.8278 | 29.84 | 0.8323 | 29.82 | 0.8323 | 29.86 | 0.8287 | 29.89 | 0.8300 | 29.89 | 0.8295 | 29.92 | 0.8360 |
| | 20 | 30.07 | 0.8510 | 31.59 | 0.8800 | 32.17 | 0.8899 | 32.23 | 0.8949 | 32.22 | 0.8930 | 32.25 | 0.8909 | - | - | 32.30 | 0.8913 | 32.28 | 0.8950 |
| | 30 | 31.41 | 0.8850 | 32.98 | 0.9090 | 33.59 | 0.9166 | 33.65 | 0.9213 | 33.65 | 0.9190 | 33.69 | 0.9174 | 33.71 | 0.9178 | 33.73 | 0.9177 | 33.69 | 0.9201 |
| | 40 | 32.35 | 0.9040 | 33.96 | 0.9250 | 34.58 | 0.9312 | 34.65 | 0.9329 | 34.64 | 0.9331 | 34.67 | 0.9317 | 34.70 | 0.9322 | 34.72 | 0.9320 | 34.67 | 0.9345 |
| Urban100 | 10 | 26.33 | 0.7816 | 28.54 | 0.8484 | 30.31 | 0.8745 | 30.81 | 0.8885 | 30.70 | 0.8875 | 30.55 | 0.8835 | 30.87 | 0.8894 | 30.81 | 0.8866 | 31.15 | 0.8941 |
| | 20 | 28.57 | 0.8545 | 31.01 | 0.9050 | 32.81 | 0.9241 | 33.33 | 0.9266 | 33.24 | 0.9270 | 33.12 | 0.9190 | - | - | 33.38 | 0.9269 | 33.51 | 0.9272 |
| | 30 | 30.00 | 0.9013 | 32.47 | 0.9312 | 34.23 | 0.9414 | 34.74 | 0.9446 | 34.67 | 0.9430 | 34.58 | 0.9417 | 34.81 | 0.9442 | 34.81 | 0.9449 | 34.84 | 0.9462 |
| | 40 | 31.06 | 0.9215 | 33.49 | 0.9412 | 35.20 | 0.9547 | 35.69 | 0.9447 | 35.62 | 0.9519 | 35.50 | 0.9515 | 35.73 | 0.9553 | 35.73 | 0.9511 | 35.75 | 0.9550 |

Table 2: *Color image JPEG compression artifact removal* results. †A single model is trained to handle multiple quality factors.

| Set | QF | JPEG PSNR | JPEG SSIM | †QGAC PSNR | †QGAC SSIM | †FBCNN PSNR | †FBCNN SSIM | †DRUNet PSNR | †DRUNet SSIM | †KGT (Ours) PSNR | †KGT (Ours) SSIM | SwinIR PSNR | SwinIR SSIM | GRL-S PSNR | GRL-S SSIM | KGT (Ours) PSNR | KGT (Ours) SSIM |
|---|---|---|---|---|---|---|---|---|---|---|---|---|---|---|---|---|---|
| LIVE1 | 10 | 25.69 | 0.7430 | 27.62 | 0.8040 | 27.77 | 0.8030 | 27.47 | 0.8045 | 28.19 | 0.8146 | 28.06 | 0.8129 | 28.13 | 0.8139 | 28.31 | 0.8176 |
| | 20 | 28.06 | 0.8260 | 29.88 | 0.8680 | 30.11 | 0.8680 | 30.29 | 0.8743 | 30.53 | 0.8781 | 30.44 | 0.8768 | 30.49 | 0.8776 | 30.61 | 0.8792 |
| | 30 | 29.37 | 0.8610 | 31.17 | 0.8960 | 31.43 | 0.8970 | 31.64 | 0.9020 | 31.89 | 0.9051 | 31.81 | 0.9040 | 31.85 | 0.9045 | 31.94 | 0.9058 |
| | 40 | 30.28 | 0.8820 | 32.05 | 0.9120 | 32.34 | 0.9130 | 32.56 | 0.9174 | 32.81 | 0.9201 | 32.75 | 0.9193 | 32.79 | 0.9195 | 32.85 | 0.9204 |
| BSDS500 | 10 | 25.84 | 0.7410 | 27.74 | 0.8020 | 27.85 | 0.7990 | 27.62 | 0.8001 | 28.25 | 0.8076 | 28.22 | 0.8075 | 28.26 | 0.8083 | 28.37 | 0.8102 |
| | 20 | 28.21 | 0.8270 | 30.01 | 0.8690 | 30.14 | 0.8670 | 30.39 | 0.8711 | 30.55 | 0.8738 | 30.54 | 0.8739 | 30.57 | 0.8746 | 30.63 | 0.8750 |
| | 30 | 29.57 | 0.8650 | 31.330 | 0.8980 | 31.45 | 0.8970 | 31.73 | 0.9003 | 31.90 | 0.9026 | 31.90 | 0.9025 | 31.92 | 0.9030 | 31.96 | 0.9035 |
| | 40 | 30.52 | 0.8870 | 32.25 | 0.9150 | 32.36 | 0.9130 | 32.66 | 0.9168 | 32.84 | 0.9190 | 32.84 | 0.9189 | 32.86 | 0.9192 | 32.88 | 0.9193 |

Table 3: GPU memory footprint of different implementations of the key-graph attention block. $N$ is the number of tokens and $K$ is the number of nearest neighbors.

| $N$ | Triton | Torch-Gather | Torch-Mask |
|---|---|---|---|
| 512 | 0.27 GB | 0.66 GB | 0.36 GB |
| 1024 | 0.33 GB | 1.10 GB | 0.67 GB |
| 2048 | 0.68 GB | 2.08 GB | 1.91 GB |
| 4096 | 2.61 GB | 4.41 GB | 6.83 GB |
| 8192 | 10.21 GB | 10.57 GB | 26.42 GB |

| $K$ | Triton | Torch-Gather | Torch-Mask |
|---|---|---|---|
| 32 | 5.51 GB | 15.00 GB | 13.68 GB |
| 64 | 5.82 GB | 27.56 GB | 13.93 GB |
| 128 | 6.45 GB | OOM | 14.43 GB |
| 256 | 7.70 GB | OOM | 15.43 GB |
| 512 | 10.20 GB | OOM | 17.43 GB |

Table 4: *Single-image motion deblurring* results. GoPro dataset Nah et al. (2017) is used for training.

| Method | GoPro PSNR↑ | GoPro SSIM↑ | HIDE PSNR↑ | HIDE SSIM↑ | Average PSNR↑ | Average SSIM↑ |
|---|---|---|---|---|---|---|
| DeblurGAN Kupyn et al. (2018) | 28.70 | 0.858 | 24.51 | 0.871 | 26.61 | 0.865 |
| Nah *et al.* Nah et al. (2017) | 29.08 | 0.914 | 25.73 | 0.874 | 27.41 | 0.894 |
| DeblurGAN-v2 Kupyn et al. (2019) | 29.55 | 0.934 | 26.61 | 0.875 | 28.08 | 0.905 |
| SRN Tao et al. (2018) | 30.26 | 0.934 | 28.36 | 0.915 | 29.31 | 0.925 |
| Gao *et al.* Gao et al. (2019) | 30.90 | 0.935 | 29.11 | 0.913 | 30.01 | 0.924 |
| DBGAN Zhang et al. (2020) | 31.10 | 0.942 | 28.94 | 0.915 | 30.02 | 0.929 |
| MT-RNN Park et al. (2020) | 31.15 | 0.945 | 29.15 | 0.918 | 30.15 | 0.932 |
| DMPHN Zhang et al. (2019a) | 31.20 | 0.940 | 29.09 | 0.924 | 30.15 | 0.932 |
| Suin *et al.* Suin et al. (2020) | 31.85 | 0.948 | 29.98 | 0.930 | 30.92 | 0.939 |
| CODE Zhao et al. (2023) | 31.94 | - | 29.67 | - | 30.81 | - |
| SPAIR Purohit et al. (2021) | 32.06 | 0.953 | 30.29 | 0.931 | 31.18 | 0.942 |
| MIMO-UNet+ Cho et al. (2021) | 32.45 | 0.957 | 29.99 | 0.930 | 31.22 | 0.944 |
| IPT Chen et al. (2021a) | 32.52 | - | - | - | - | - |
| MPRNet Zamir et al. (2021) | 32.66 | 0.959 | 30.96 | 0.939 | 31.81 | 0.949 |
| KiT Lee et al. (2022) | 32.70 | 0.959 | 30.98 | 0.942 | 31.84 | 0.951 |
| Restormer Zamir et al. (2022) | 32.92 | 0.961 | 31.22 | 0.942 | 32.07 | 0.952 |
| Ren *et al.* Ren et al. (2023b) | 33.20 | 0.963 | 30.96 | 0.938 | 32.08 | 0.951 |
| KGT (ours) | 33.44 | 0.964 | 31.05 | 0.941 | 32.25 | 0.953 |

optimize the efficiency of the proposed KGT, we recommend employing *Torch-Mask* during training and *Triton* during inference, striking a balance between the efficiency and the GPU memory requirement.

## 4.2 EVALUATION OF KGT ON VARIOUS IR TASKS

**Evaluation on JPEG Compression Artifact Reduction.** For JPEG CAR, the experiments for both grayscale and color images are conducted with four image quality factors ranging from 10 to 40 under two experimental settings (*i.e.,* a single model is trained to handle multiple quality factors, and each model for each image quality). The quantitative results shown in Tab. 1 validate that the KGT outperforms all other methods like DnCNN-3 Zhang et al. (2017), DRUNet Zhang et al. (2021), GRL-S Li et al. (2023a), SwinIR Liang et al. (2021), ART Zhang et al. (2023b), and CAT Chen et al. (2022b) under both settings. Besides, the results for color images in Tab. 2 also show that our KGT achieves the best results on all the test sets and quality factors among all compared methods like QGAC Ehrlich et al. (2020), FBCNN Jiang et al. (2021), DRUNet, SwinIR, and GRL-S. The visual comparisons in the *Supp. Mat.* further support the effectiveness of the proposed KGT.

Table 5: *Color and grayscale image denoising* results. Model complexity and prediction accuracy are shown for better comparison. †A single model is trained to handle multiple noise levels.

| Method | # P | Color | | | | | | | | | Grayscale | | | | | | | | |
|---|---|---|---|---|---|---|---|---|---|---|---|---|---|---|---|---|---|---|---|
| | | CBSD68 | | | McMaster | | | Urban100 | | | Set12 | | | BSD68 | | | Urban100 | | |
| | | $\sigma$=15 | $\sigma$=25 | $\sigma$=50 | $\sigma$=15 | $\sigma$=25 | $\sigma$=50 | $\sigma$=15 | $\sigma$=25 | $\sigma$=50 | $\sigma$=15 | $\sigma$=25 | $\sigma$=50 | $\sigma$=15 | $\sigma$=25 | $\sigma$=50 | $\sigma$=15 | $\sigma$=25 | $\sigma$=50 |
| †DnCNN | 0.56 | 33.90 | 31.24 | 27.95 | 33.45 | 31.52 | 28.62 | 32.98 | 30.81 | 27.59 | 32.67 | 30.35 | 27.18 | 31.62 | 29.16 | 26.23 | 32.28 | 29.80 | 26.35 |
| †FFDNet | 0.49 | 33.87 | 31.21 | 27.96 | 34.66 | 32.35 | 29.18 | 33.83 | 31.40 | 28.05 | 32.75 | 30.43 | 27.32 | 31.63 | 29.19 | 26.29 | 32.40 | 29.90 | 26.50 |
| †DRUNet | 32.64 | 34.30 | 31.69 | 28.51 | 35.40 | 33.14 | 30.08 | 34.81 | 32.60 | 29.61 | 33.25 | 30.94 | 27.90 | 31.91 | 29.48 | 26.59 | 33.44 | 31.11 | 27.96 |
| †Restormer | 26.13 | 34.39 | 31.78 | 28.59 | 35.55 | 33.31 | 30.29 | 35.06 | 32.91 | 30.02 | 33.35 | 31.04 | 28.01 | 31.95 | 29.51 | 26.62 | 33.67 | 31.39 | 28.33 |
| †KGT (Ours) | 25.82 | 34.42 | 31.78 | 28.57 | 35.65 | 33.40 | 30.34 | 35.37 | 33.26 | 30.41 | 33.47 | 31.16 | 28.12 | 31.95 | 29.49 | 26.54 | 34.05 | 31.84 | 28.83 |
| DnCNN | 0.56 | 33.90 | 31.24 | 27.95 | 33.45 | 31.52 | 28.62 | 32.98 | 30.81 | 27.59 | 32.86 | 30.44 | 27.18 | 31.73 | 29.23 | 26.23 | 32.64 | 29.95 | 26.26 |
| RNAN | 8.96 | - | - | 28.27 | - | - | 29.72 | - | - | 29.08 | - | - | 27.70 | - | - | 26.48 | - | - | 27.65 |
| IPT | 115.33 | - | - | 28.39 | - | - | 29.98 | - | - | 29.71 | - | - | - | - | - | - | - | - | - |
| EDT-B | 11.48 | 34.39 | 31.76 | 28.56 | 35.61 | 33.34 | 30.25 | 35.22 | 33.07 | 30.16 | - | - | - | - | - | - | - | - | - |
| DRUNet | 32.64 | 34.30 | 31.69 | 28.51 | 35.40 | 33.14 | 30.08 | 34.81 | 32.60 | 29.61 | 33.25 | 30.94 | 27.90 | 31.91 | 29.48 | 26.59 | 33.44 | 31.11 | 27.96 |
| SwinIR | 11.75 | 34.42 | 31.78 | 28.56 | 35.61 | 33.20 | 30.22 | 35.13 | 32.90 | 29.82 | 33.36 | 31.01 | 27.91 | 31.97 | 29.50 | 26.58 | 33.70 | 31.30 | 27.98 |
| Restormer | 26.13 | 34.40 | 31.79 | 28.60 | 35.61 | 33.34 | 30.30 | 35.13 | 32.96 | 30.02 | 33.42 | 31.08 | 28.00 | 31.96 | 29.52 | 26.62 | 33.79 | 31.46 | 28.29 |
| Xformer | 25.23 | 34.43 | 31.82 | 28.63 | 35.68 | 33.44 | 30.38 | 35.29 | 33.21 | 30.36 | 33.46 | 31.16 | 28.10 | 31.98 | 29.55 | 26.65 | 33.98 | 31.78 | 28.71 |
| KGT (Ours) | 25.82 | 34.43 | 31.79 | 28.60 | 35.65 | 33.43 | 30.38 | 35.38 | 33.29 | 30.51 | 33.48 | 31.18 | 28.14 | 31.97 | 29.52 | 26.53 | 34.09 | 31.87 | 28.86 |

Table 6: *Image Restoration in adverse weather conditions*.  Table 7: *Image demosaicking* results.

| Type | Test1 (rain+fog) | | SnowTest100k-L | | RainDrop | |
|---|---|---|---|---|---|---|
| | Method | PSNR | Method | PSNR | Method | PSNR |
| Task Specific | pix2pix | 19.09 | DesnowNet | 27.17 | Attn. GAN | 30.55 |
| | HRGAN | 21.56 | JSTASR | 25.32 | Quan *et al.* | 31.44 |
| | SwinIR | 23.23 | SwinIR | 28.18 | SwinIR | 30.82 |
| | MPRNet | 21.90 | DDMSNET | 28.85 | CCN | 31.34 |
| Multi Task | All-in-One | 24.71 | All-in-One | 28.33 | All-in-One | 31.12 |
| | TransWeather | 27.96 | TransWeather | 28.48 | TransWeather | 28.84 |
| | KGT (Ours) | 29.57 | KGT (Ours) | 30.76 | KGT (Ours) | 30.82 |

| Datasets | Kodak | McMaster |
|---|---|---|
| Matlab | 35.78 | 34.43 |
| MMNet Kokkinos et al. (2019) | 40.19 | 37.09 |
| DDR Wu et al. (2016) | 41.11 | 37.12 |
| DeepJoint Gharbi et al. (2016) | 42.00 | 39.14 |
| RLDD Guo et al. (2020) | 42.49 | 39.25 |
| DRUNet Zhang et al. (2021) | 42.68 | 39.39 |
| RNAN Zhang et al. (2019b) | 43.16 | 39.70 |
| GRL Li et al. (2023a) | 43.57 | 40.22 |
| KGT (Ours) | 43.62 | 40.68 |

**Evaluation on Image Denoising.** We show color and grayscale image denoising results in Tab. 5 under two settings (*i.e.*, † one model for all noise levels $\sigma = \{15, 25, 50\}$ and each model for each noise level). For a fair comparison, both the network complexity and accuracy are reported for all the comparison methods like DnCNN Zhang et al. (2017), FFDNet Zhang et al. (2018a), DRUNet Zhang et al. (2021), RNAN Zhang et al. (2019b), IPT Chen et al. (2021a), EDT Li et al. (2021a), SwinIR Liang et al. (2021), Restormer Zamir et al. (2022), and Xformer Zhang et al. (2023a). For setting †, our KGT achieves better performance on all test datasets for both color and grayscale image denoising compared to all others. It's worth noting that we outperform DRUNet and Restormer with lower trainable parameters. For another setting, our KGT also archives better results on CBSD68 and Urban100 for color image denoising and on Set12 and Urban100 for grayscale denoising. These interesting comparisons validate the effectiveness of the proposed KGT and also indicate that KGT may have a higher generalization. The qualitative results in *Supp. Mat.* also supports that the proposed KGT can remove heavy noise corruption and preserve high-frequency image details.

**Evaluation on Image Demosaicking.** The quantitative results are shown in 7, which show that the proposed KGT archives the best performance on both the Kodak and MaMaster test datasets. Especially, 0.05dB and 0.45dB absolute improvement compared to the current state-of-the-art GRL.

**Evaluation in Adverse Weather Conditions.** We validate KGT in adverse weather conditions like rain+fog (Test1), snow (SnowTest100K), and raindrops (RainDrop). The PSNR score is reported in Tab. 6 for each method. It's clear that our method achieves the best performance on Test1 (*i.e.*, 5.76% improvement) and SnowTest100k-L (*i.e.* 8.01% improvement), while the second-best PSNR on RainDrop compared to all other methods. The visual comparison in our *Supp. Mat.*.

**Evaluation on SR.** For the classical image SR, we compared our KGT with both recent lightweight and accurate SR models, and the quantitative results are shown in Tab. 8. Compared to EDT, KGT-base achieves significant improvements on Urban100 (*i.e.*, 0.72 dB and 0.76dB for x2 and x4 SR) and Manga109 datasets (*i.e.*, 0.22dB and 0.17 dB for x2 and x4 SR). Furthermore, even the KGT-small consistently ranks as the runner-up in terms of performance across the majority of test datasets, all while maintaining a reduced number of trainable parameters. The visual results shown in Fig. 5 also validate the effectiveness of the proposed KGT in restoring more details and structural content.

**Evaluation on Image deblurring.** Tab. 4 shows the quantitative results for single image motion deblurring on synthetic datasets (GoPro Nah et al. (2017), HIDE Shen et al. (2019)). Compared to

Table 8: *Classical image SR* results. Both lightweight and accurate models are summarized.

| Method | Scale | Params [M] | Set5 PSNR | Set5 SSIM | Set14 PSNR | Set14 SSIM | BSD100 PSNR | BSD100 SSIM | Urban100 PSNR | Urban100 SSIM | Manga109 PSNR | Manga109 SSIM |
|---|---|---|---|---|---|---|---|---|---|---|---|---|
| RCAN Zhang et al. (2018b) | ×2 | 15.44 | 38.27 | 0.9614 | 34.12 | 0.9216 | 32.41 | 0.9027 | 33.34 | 0.9384 | 39.44 | 0.9786 |
| SAN Dai et al. (2019) | ×2 | 15.71 | 38.31 | 0.9620 | 34.07 | 0.9213 | 32.42 | 0.9028 | 33.10 | 0.9370 | 39.32 | 0.9792 |
| HAN Niu et al. (2020) | ×2 | 63.61 | 38.27 | 0.9614 | 34.16 | 0.9217 | 32.41 | 0.9027 | 33.35 | 0.9385 | 39.46 | 0.9785 |
| IPT Chen et al. (2021a) | ×2 | 115.48 | 38.37 | - | 34.43 | - | 32.48 | - | 33.76 | - | - | - |
| SwinIR Liang et al. (2021) | ×2 | 11.75 | 38.42 | 0.9623 | 34.46 | 0.9250 | 32.53 | 0.9041 | 33.81 | 0.9427 | 39.92 | 0.9797 |
| CAT-A (Chen et al., 2022b) | ×2 | 16.46 | 38.51 | 0.9626 | 34.78 | 0.9265 | 32.59 | 0.9047 | 34.26 | 0.9440 | 40.10 | 0.9805 |
| ART Zhang et al. (2023b) | ×2 | 16.40 | 38.56 | 0.9629 | 34.59 | 0.9267 | 32.58 | 0.9048 | 34.30 | 0.9452 | 40.24 | 0.9808 |
| EDT Li et al. (2021a) | ×2 | 11.48 | 38.63 | 0.9632 | 34.80 | 0.9273 | 32.62 | 0.9052 | 34.27 | 0.9456 | 40.37 | 0.9811 |
| KGT-S (Ours) | ×2 | 11.87 | 38.57 | 0.9651 | 34.99 | 0.9300 | 32.65 | 0.9078 | 34.86 | 0.9472 | 40.45 | 0.9824 |
| KGT-B (Ours) | ×2 | 19.90 | 38.61 | 0.9654 | 35.08 | 0.9304 | 32.69 | 0.9084 | 34.99 | 0.9455 | 40.59 | 0.9830 |
| RCAN Zhang et al. (2018b) | ×3 | 15.63 | 34.74 | 0.9299 | 30.65 | 0.8482 | 29.32 | 0.8111 | 29.09 | 0.8702 | 34.44 | 0.9499 |
| SAN Dai et al. (2019) | ×3 | 15.90 | 34.75 | 0.9300 | 30.59 | 0.8476 | 29.33 | 0.8112 | 28.93 | 0.8671 | 34.30 | 0.9494 |
| HAN Niu et al. (2020) | ×3 | 64.35 | 34.75 | 0.9299 | 30.67 | 0.8483 | 29.32 | 0.8110 | 29.10 | 0.8705 | 34.48 | 0.9500 |
| NLSA Mei et al. (2021) | ×3 | 45.58 | 34.85 | 0.9306 | 30.70 | 0.8485 | 29.34 | 0.8117 | 29.25 | 0.8726 | 34.57 | 0.9508 |
| IPT Chen et al. (2021a) | ×3 | 115.67 | 34.81 | - | 30.85 | - | 29.38 | - | 29.49 | - | - | - |
| SwinIR Liang et al. (2021) | ×3 | 11.94 | 34.97 | 0.9318 | 30.93 | 0.8534 | 29.46 | 0.8145 | 29.75 | 0.8826 | 35.12 | 0.9537 |
| CAT-A (Chen et al., 2022b) | ×3 | 16.64 | 35.06 | 0.9326 | 31.04 | 0.8538 | 29.52 | 0.8160 | 30.12 | 0.8862 | 35.38 | 0.9546 |
| ART Zhang et al. (2023b) | ×3 | 16.58 | 35.07 | 0.9325 | 31.02 | 0.8541 | 29.51 | 0.8159 | 30.10 | 0.8871 | 35.39 | 0.9548 |
| EDT Li et al. (2021a) | ×3 | 11.66 | 35.13 | 0.9328 | 31.09 | 0.8553 | 29.53 | 0.8165 | 30.07 | 0.8863 | 35.47 | 0.9550 |
| KGT-S (Ours) | ×3 | 12.05 | 34.99 | 0.9366 | 31.23 | 0.8594 | 29.53 | 0.8223 | 30.71 | 0.8950 | 35.52 | 0.9573 |
| KGT-B (Ours) | ×3 | 20.08 | 35.03 | 0.9371 | 31.29 | 0.8603 | 29.54 | 0.8227 | 30.87 | 0.9012 | 35.60 | 0.9581 |
| RCAN Zhang et al. (2018b) | ×4 | 15.59 | 32.63 | 0.9002 | 28.87 | 0.7889 | 27.77 | 0.7436 | 26.82 | 0.8087 | 31.22 | 0.9173 |
| SAN Dai et al. (2019) | ×4 | 15.86 | 32.64 | 0.9003 | 28.92 | 0.7888 | 27.78 | 0.7436 | 26.79 | 0.8068 | 31.18 | 0.9169 |
| HAN Niu et al. (2020) | ×4 | 64.20 | 32.64 | 0.9002 | 28.90 | 0.7890 | 27.80 | 0.7442 | 26.85 | 0.8094 | 31.42 | 0.9177 |
| IPT Chen et al. (2021a) | ×4 | 115.63 | 32.64 | - | 29.01 | - | 27.82 | - | 27.26 | - | - | - |
| SwinIR Liang et al. (2021) | ×4 | 11.90 | 32.92 | 0.9044 | 29.09 | 0.7950 | 27.92 | 0.7489 | 27.45 | 0.8254 | 32.03 | 0.9260 |
| CAT-A (Chen et al., 2022b) | ×4 | 16.60 | 33.08 | 0.9052 | 29.18 | 0.7960 | 27.99 | 0.7510 | 27.89 | 0.8339 | 32.39 | 0.9285 |
| ART Zhang et al. (2023b) | ×4 | 16.55 | 33.04 | 0.9051 | 29.16 | 0.7958 | 27.97 | 0.751 | 27.77 | 0.8321 | 32.31 | 0.9283 |
| EDT Li et al. (2021a) | ×4 | 11.63 | 33.06 | 0.9055 | 29.23 | 0.7971 | 27.99 | 0.7510 | 27.75 | 0.8317 | 32.39 | 0.9283 |
| KGT-S (Ours) | ×4 | 12.02 | 33.02 | 0.9082 | 29.29 | 0.8026 | 27.96 | 0.7582 | 28.34 | 0.8467 | 32.48 | 0.9322 |
| KGT-B (Ours) | ×4 | 20.04 | 33.08 | 0.9090 | 29.34 | 0.8037 | 27.98 | 0.7599 | 28.51 | 0.8467 | 32.56 | 0.9335 |

Figure 5: Visual comparison of classical image SR (x4) on Urban100. Best viewed by zooming.

the previous state-of-the-art Restormer Zamir et al. (2022), the proposed KGT achieves significant PSNR improvement of 0.52 dB on the GoPro dataset and the second-best performance with a slightly lower PSNR and SSIM on HIDE dataset. More visual results are shown in the *Supp. Mat.*.

## 5 CONCLUSION

In this paper, for the first time, we utilize ViTs from the graph perspective specifically tailored for image restoration with the proposed KGT for both the widely-used multi-stage (for image SR) and the U-shaped architectures (For other IR tasks). In particular, a key graph is constructed that can capture the complex relation of each node feature with only the most relevant K nodes with the proposed Key-Graph constructor instead of a dense fully connected graph. Then the Key-Graph is shared with all the KGT layers within the same KGT stage, which enables the Key-Graph Attention block to capture fewer but the Key relation of each node. As a result, KGT leads to the computation complexity reduced from $\mathcal{O}((hw)^2)$ to $\mathcal{O}((hw) \times K)$, which largely released the potential of ViTs in a sparse yet representative manner. Extensive experiments on 6 IR tasks validated the effectiveness of the proposed KGT and the results demonstrate that the proposed KGT achieves new state-of-the-art performance. The code of the proposed KGT will be released.

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
