# KEY-GRAPH TRANSFORMER FOR IMAGE RESTORATION: SUPPLEMENTARY MATERIAL

## 1 EXPERIMENTAL REPORTING

### 1.1 MODEL ARCHITECTURE

In the proposed KGT, we adopt two kinds of base architecture *i.e.,* the widely used multi-stage one shown in Fig.1 of our main manuscript (Archi-V1) and a U-shaped hierarchical one shown in Fig. 1 (Archi-V2) for taking patterns of various scales into account (Note that 1/3 of $I_{low}$ and $I_{high}$ in Fig. 1 denotes the grayscale/color image cases). This is consistent with previous methods such as Restormer Zamir et al. (2022), KiT Lee et al. (2022), and NAFNet Chen et al. (2022). The rationale behind adopting the U-shape structure for tasks other than super-resolution is its computational efficiency, particularly in maintaining restoration at the original image size level. This strategic choice ensures that our model remains efficient while addressing diverse tasks. The resulting 25.82 Mb trainable parameter count strikes a balance between the model size of SwinIR Liang et al. (2021) and Restormer Zamir et al. (2022).

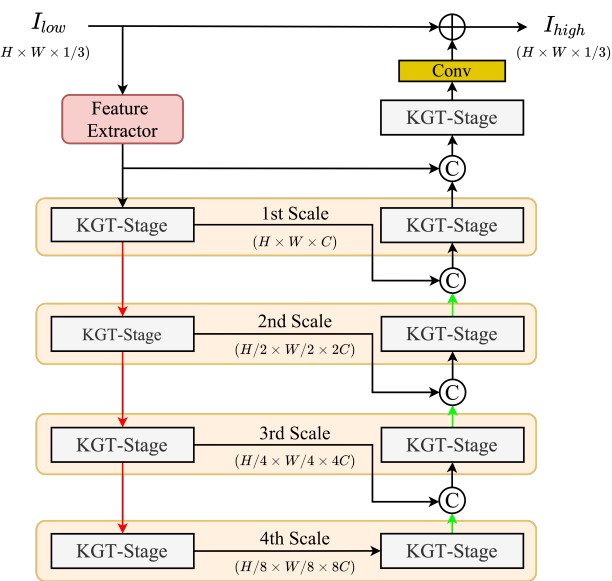

Figure 1: The U-shaped hierarchical architecture (Archi-V2) of the proposed Key-Graph Transformer (KGT) for Image Restoration. Note that this U-shaped one is used for image JPEG CAR, image denoising, image demosaicking, IR in AWC, and image deblurring. Symbols $\oplus$ and $\copyright$ denote the element-wise addition and channel-wise concatenation. The downsample and upsample operations are denoted by red and green arrows.

Table 1: The details of the KGT stages and KGT layers per stage for both architectures.

|  | Archi-V1 | | Archi-V2 | | |
| --- | --- | --- | --- | --- | --- |
|  | KGT-small | KGT-base | Down Stages | Up Stages | Final Stage |
| Num. of KGT Stages | 6 | 8 | 4 | 4 | 1 |
| Num. of KGT layer/stage | 6 | 8 | 6 | 6 | 6 |

Table 2: The training details of the KGT for various IR tasks.

| | | | JPEG CAR | Denoising | Demosaicking | IR in AWC | SR | Deblurring |
|---|---|---|---|---|---|---|---|---|
| | Architecture | | Archi-V2 | Archi-V2 | Archi-V2 | Archi-V2 | Archi-V1 | Archi-V2 |
| Training | Phase1 | Dataset | ImageNet | ImageNet | ImageNet | All-Weather | ImageNet | GoPro |
| | (Pre-Training) | Iters | 1M | 1M | 1M | - | 1M | 450K |
| | Phase2 | Iters | 1M | 1M | 1M | 750K | 1M | 200K |
| | Optimizer | | AdamW | AdamW | AdamW | Adam | AdamW | AdamW |
| | Loss Function | | L1 | L1 | L1 | smooth L1, VGG | L1 | Charbonnier |
| Warmup | Iterations | | 50K | 50K | 50K | 50K | 50K | 50K |
| | Init lr | | 1e-5 | 1e-5 | 1e-5 | 1e-5 | 1e-5 | 1e-5 |
| | End lr | | 2e-4 | 2e-4 | 2e-4 | 2e-4 | 2e-4 | 2e-4 |
| lr | Init lr | | 2e-4 | 2e-4 | 2e-4 | 2e-4 | 2e-4 | 2e-4 |
| | Decay Type | | half decay | half decay | half decay | half decay | half decay | half decay |
| | Milestone | | [400, 700, 850, 900, 950]K | =JPEG CAR | =JPEG CAR | [200, 300, 350, 400, 450]K | =JPEG CAR | [50, 100, 125, 150]K |
| | Batch size | | 16 | 16 | 16 | 32 | 16 | 8 |
| | Patch size | | 64 | 64 | 64 | 16 | 64 | 192 |

In addition to introducing the two base architectures of the proposed KGT, we have included comprehensive details on the structure of the KGT in Table 1. This table outlines the number of KGT stages and the distribution of KGT layers within each stage for a more thorough understanding of our model's architecture.

## 1.2 TRAINING DETAILS

Our method explores six different IR tasks, and the training settings vary slightly for each task. These differences encompass the architecture of the proposed KGT, variations in training phases, choice of the optimizer, employed loss functions, warm-up settings, learning rate schedules, batch sizes, and patch sizes. We have provided a comprehensive overview of these details in Tab. 2.

In addition, there are several points about the training details we want to make further explanation. 1) The two-phase training strategy is conducted by pre-training on ImageNet Deng et al. (2009) together with task-specific training on the corresponding training datasets. This is inspired by previous works Dong et al. (2014); Chen et al. (2021); Li et al. (2021); Chen et al. (2023). 2) For IR in AWC, we adopted similar training settings as Transweather Valanarasu et al. (2022), the model is trained for a total of 750K iterations (Iters) without pre-training. We trained the model for JPEG CAR, denoising, demosaicking, and SR for a total of 1M Iters in the second phase with pre-training for 1M Iters in the first phase. For deblurring, we trained the model for 200K Iters in the second phase with a pre-training of 450K Iters in the first phase. 3) The optimizer used for IR in AWC is Adam Kingma & Ba (2014), while AdamW Loshchilov & Hutter (2018) is used for the rest IR tasks. 4) The training losses for IR in AWC are the smooth L1 and the Perception VGG loss Johnson et al. (2016); Simonyan & Zisserman (2015). For image deblurring, the training loss is the Charbonnier loss. For the rest IR task, the L1 loss is commonly used for the training.

## 1.3 TRAINING/TESTING DATASETS

For JPEG CAR, image denoising, image demosaicking, image SR, and image deblurring, we all adopted the two-phase training strategies. Since the first phase is all pre-training on the ImageNet. We only report the training dataset of the second phase and the corresponding test datasets. For IR in AWC, we used a similar training pipeline as Transweather with only one phase. Additionally, for tasks such as JPEG CAR, image denoising, demosaicking, and image super-resolution (SR), how low-quality images are generated is also briefly introduced below.

**JPEG compression artifact removal.** For JPEG compression artifact removal, the JPEG image is compressed by the `cv2` JPEG compression function. The compression function is characterized by the quality factor. We investigated four compression quality factors including 10, 20, 30, and 40. The smaller the quality factor, the more the image is compressed, meaning a lower quality.

- The training datasets: DIV2K Agustsson & Timofte (2017), Flickr2K Lim et al. (2017), and WED Ma et al. (2016).

- The test datasets: Classic5 Foi et al. (2007), LIVE1 Sheikh (2005), Urban100 Huang et al. (2015), BSD500 Arbelaez et al. (2010).

**Image Denoising.** For image denoising, we conduct experiments on both color and grayscale image denoising. During training and testing, noisy images are generated by adding independent additive white Gaussian noise (AWGN) to the original images. The noise levels are set to $\sigma = 15, 25, 50$. We train individual networks at different noise levels. The network takes the noisy images as input and tries to predict noise-free images.

- The training datasets: DIV2K Agustsson & Timofte (2017), Flickr2K Lim et al. (2017), WED Ma et al. (2016), and BSD400 Martin et al. (2001).
- The test datasets for color image: CBSD68 Martin et al. (2001), Kodak24 Franzen (1999), McMaster Zhang et al. (2011), and Urban100 Huang et al. (2015).
- The test datasets for grayscale image: Set12 Zhang et al. (2017), BSD68 Martin et al. (2001), and Urban100 Huang et al. (2015).

**Image Demosaicking.** For image demosaicking, the mosaic image is generated by applying a Bayer filter on the ground-truth image. Then the network try to restore high-quality image. The mosaic image is first processed by the default `Matlab` demosaic function and then passed to the network as input.

- The training datasets: DIV2K Agustsson & Timofte (2017) and Flickr2K Lim et al. (2017).
- The test datasets: Kodak Franzen (1999), McMaster Zhang et al. (2011).

**IR in Adverse Weather Conditions.** For IR in adverse weather conditions, the model is trained on a combination of images degraded by a variety of adverse weather conditions. The same training and test dataset is used as in Transweather Valanarasu et al. (2022). The training data comprises 9,000 images sampled from Snow100K Liu et al. (2018), 1,069 images from Raindrop Qian et al. (2018), and 9,000 images from Outdoor-Rain Li et al. (2019). Snow100K includes synthetic images degraded by snow, Raindrop consists of real raindrop images, and Outdoor-Rain contains synthetic images degraded by both fog and rain streaks. The proposed method is tested on both synthetic and real-world datasets.

- The comparison methods in Tab. 6 of our main manuscript: pix2pix Isola et al. (2017), HRGAN Li et al. (2019), SwinIR Liang et al. (2021), All-in-One Li et al. (2020), Transweather Valanarasu et al. (2022), DesnowNet Liu et al. (2018), JSTASR Chen et al. (2020), DDMSNET Zhang et al. (2021), Attn. GAN Qian et al. (2018), Quan et al. (2019), and CC-GAN Quan et al. (2021).
- The test datasets: test1 dataset Li et al. (2020; 2019), the RainDrop test dataset Qian et al. (2018), and the Snow100k-L test.

**Image SR.** For image SR, the LR image is synthesized by `Matlab` bicubic downsampling function before the training. We investigated the upscalingg factors $\times 2$, $\times 3$, and $\times 4$.

- The training datasets: DIV2K Agustsson & Timofte (2017) and Flickr2K Lim et al. (2017).
- The test datasets: Set5 Bevilacqua et al. (2012), Set14 Zeyde et al. (2010), BSD100 Martin et al. (2001), Urban100 Huang et al. (2015), and Manga109 Matsui et al. (2017).

**Image Deblurring.** For single-image motion deblurring,

- The training datasets: GoPro Nah et al. (2017) .
- The test datasets: GoPro Nah et al. (2017) and HIDE Shen et al. (2019).

## 1.4 EVALUATION INTRODUCTION

Note that the results of all the comparison methods are reported from their original papers. The details of the evaluation metric (*i.e.,* SSIM, PSNR) are described as follows:

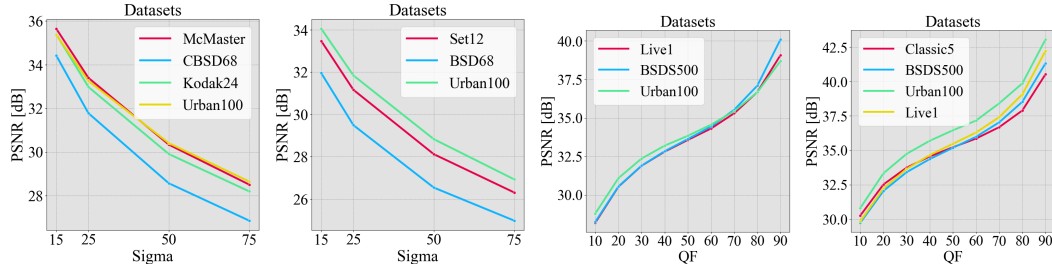

(a) Color image denoising. One model for multiple noise levels.

(b) Grayscale image denoising. One model for multiple noise levels.

(c) Color image JPEG CAR. One model for multiple quality factors.

(d) Grayscale image JPEG CAR. One model for multiple quality factors.

Figure 2: One model is trained to handle multiple degradation levels.

**JPEG compression artifact removal.** For color image JPEG compression artifact removal, the PSNR is reported on the RGB channels while for grayscale image JPEG compression artifact removal, the PSNR is reported on the Y channel.

**Image Denoising.** For color image denoising, the PSNR is reported on the RGB channels while for grayscale image denoising, the PSNR is reported on the Y channel.

**Image Demosaicking.** For the comparison between different methods, PSNR is reported on the RGB channels.

**IR in Adverse Weather Conditions.** We adopted the same PSNR evaluation metric used in Transweather Valanarasu et al. (2022).

**Image SR.** The PSNR is reported on the Y channel.

**Image Deblurring.** The PSNR and SSIM on the RGB channels are reported.

## 2 LIMITATIONS

This study faces a task-specific limitation: each image restoration task requires training a separate network. While efforts have been made to train models for varying degradation levels within specific types, such as image denoising and removal of JPEG compression artifacts, this approach still leads to inefficiencies in model training and constrains the utility of the trained networks. A potential future enhancement involves developing a mechanism enabling a network to handle diverse image degradation types and levels. Another challenge is the substantial parameter requirement of the proposed KGT, which operates within a tens-of-millions parameter budget. Deploying such a large image IR network on handheld devices with limited resources is challenging, if not unfeasible. Therefore, a promising research direction is the creation of more efficient versions of KGT, integrating non-local context more effectively, to overcome these limitations.

## 3 MORE ABLATION ANALYSES

Besides the ablation studies presented in our main manuscript, we further provide the following two analyses:

**The Impact of One Model is Trained to Handle Multiple Degradation Levels.** Training one model to tackle multiple degradation levels is conducted in our paper to validate the generalization ability of the proposed method. Specifically, we choose two IR tasks (*i.e.,* image denoising and JPEG CAR) for validating the effectiveness of the proposed KGT on various datasets for both the color image and grayscale images. For image denoising, the degraded factor Sigma is set to 15, 25, 50, and 75, respectively. For JEPG CAR, the degraded factor QF is set to 10, 20, 30, 40, 50, 60, 70, 80, and 90, respectively. The comparison results are shown in Fig. 2. It shows that: (1) The PSNR for both image denoising and JPEG CAR on all the corresponding datasets, under both color and grayscale settings, decreases when the degraded level improves. (2) The proposed KGT archives better performance on the Urban100 dataset for both the grayscale image denoising and

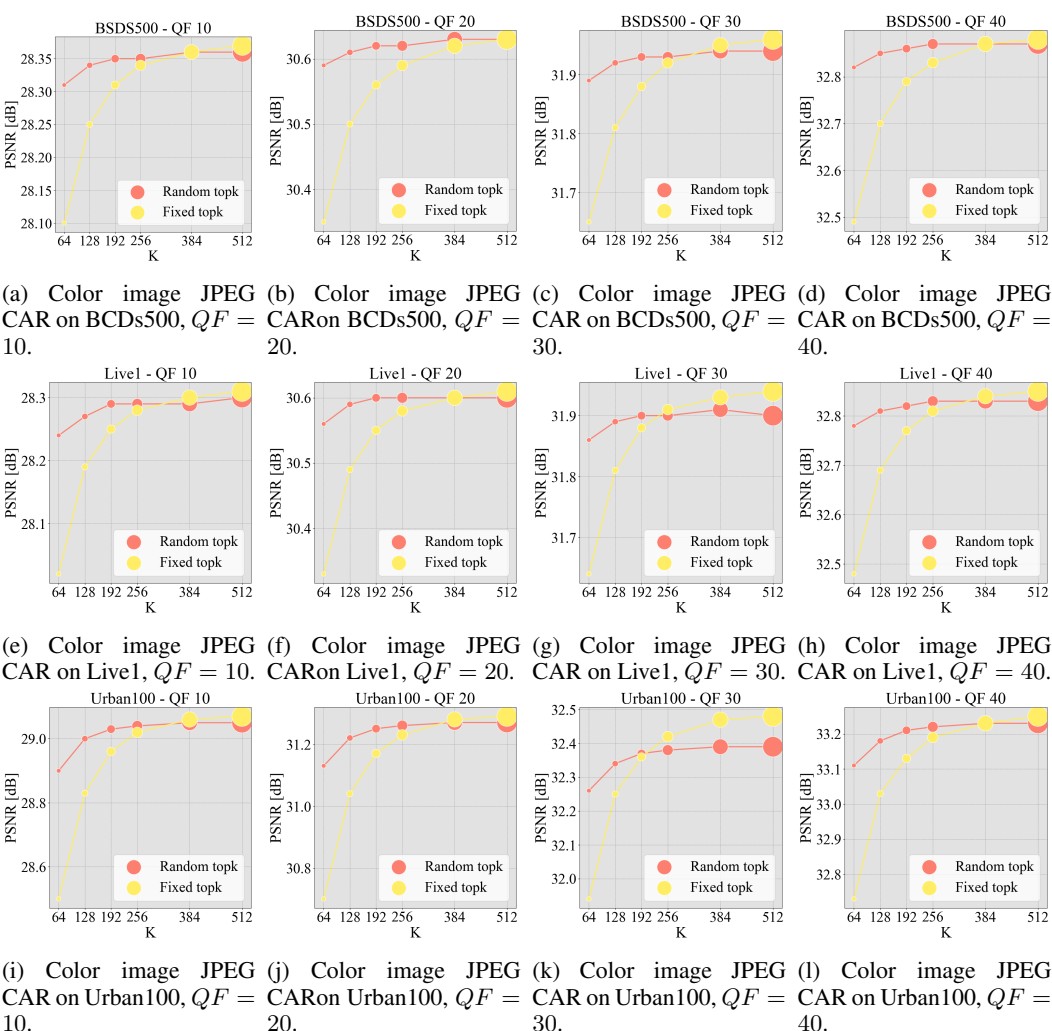

Figure 3: Ablation study on the impact of $K$ for **Color JPEG CAR** on CBSD68, Kodak24, Mc-Master, and Urban100 datasets with $QF = \{10, 20, 30, 40\}$.

the JPEG CAR tasks. Drawing from these two observations, we can conclude that training a single model to handle multiple degradation levels often results in enhanced generalization, albeit with a slight trade-off in performance compared to its counterpart, where a distinct model is trained for each degradation level.

**The Impact of the K in K-Graph Constructor under Various IR Tasks.** To explore how the K value of topK will affect the IR performance of the proposed KGT. We conduct exhaustive experiments on JPEG compression artifact reduction for both color and grayscale images under different QF values (*i.e.,* QF = [10, 20, 30, 40]), image denoising for both color and grayscale images under different noise levels (*i.e.,* Sigma = [15, 25, 50]), as well as image SR under different scale (*i.e.,* x2, x3, x4) with the proposed KGT. Note that all the experiments for each IR task are conducted under two kinds of topK settings, *i.e.,* (i) K was randomly sampled from the range [64, 512] during the overall training phase, and (ii) K was held constant at 512 throughout the training phase. For inference, K was configured to the specified value for both settings.

The results of the JPEG CAR in terms of the hyper-parameters K under different training settings during inference for both color image and grayscale image are shown in Fig. 3 and Fig. 4. It's clear that for both the color and grayscale JPEG CAR task, when K is set to 64 during inference, there is a huge performance cat between the random topK setting and the fixed topK setting. In addition, the fixed topK setting performs well or sometimes even a bit better than the random topK setting only when K is also set to the same number (*i.e.,* 512). With the decrease of K during inference for

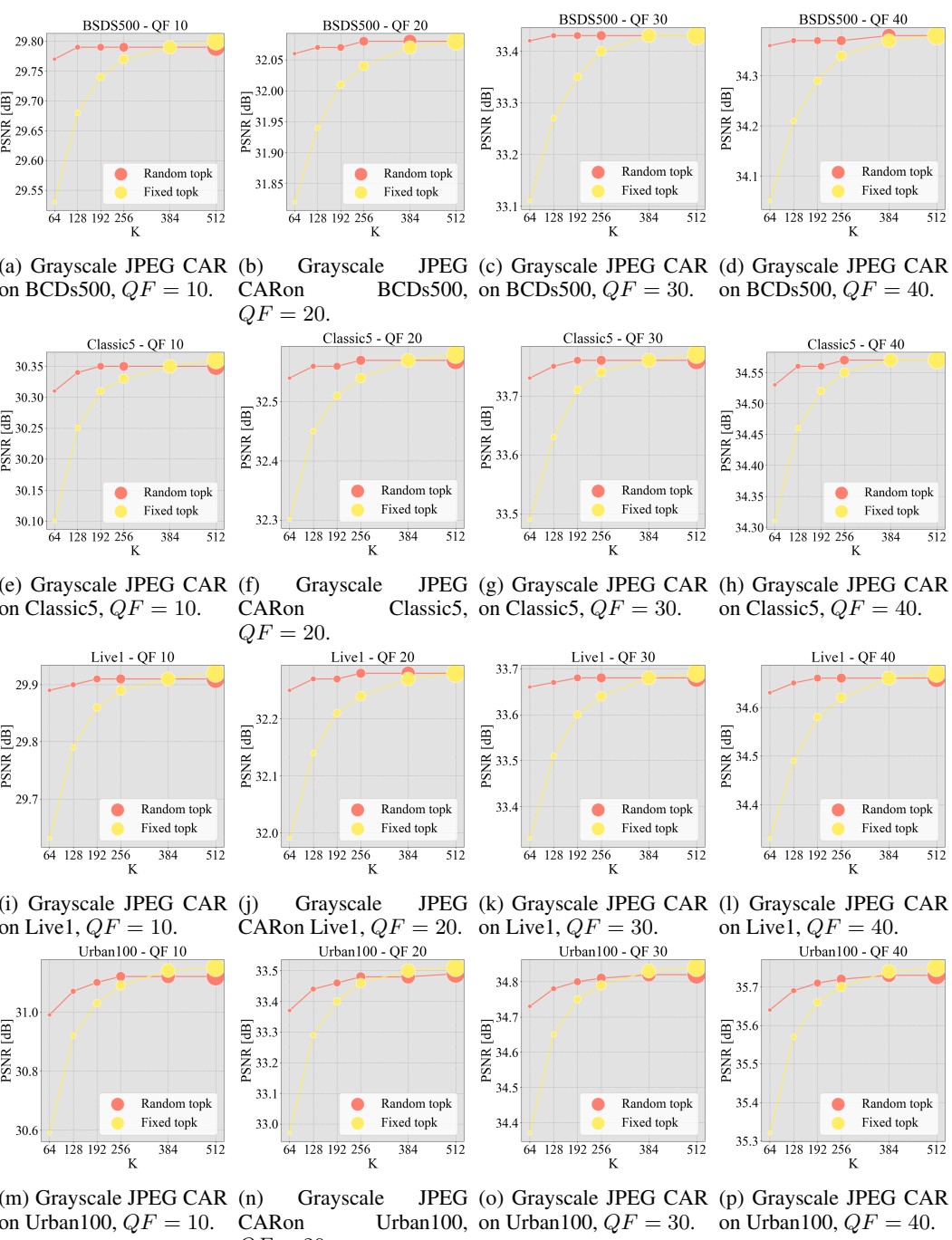

Figure 4: Ablation study on the impact of $K$ for **Grayscale JPEG CAR** on BSD68, Classic5, Live1, and Urban100 datasets with $QF = \{0, 20, 30, 40\}$.

the fixed topK setting, the PSNR drops largely marginally for all the datasets under every kind of degraded QF factor.

The results of the image denoising in terms of hyper-parameters K under different training settings during inference for both color image and grayscale image are shown in Fig. 5 and Fig. 6. All the experimental results on various datasets (*i.e.,* BSD68/CBSD68, Kodak24, McMaster, and Urban100) share a similar trend for image denoising compared to the JPEG CAR task. The random topK setting can maintain a relatively stable PSNR score under different K during inference compared to its fixed

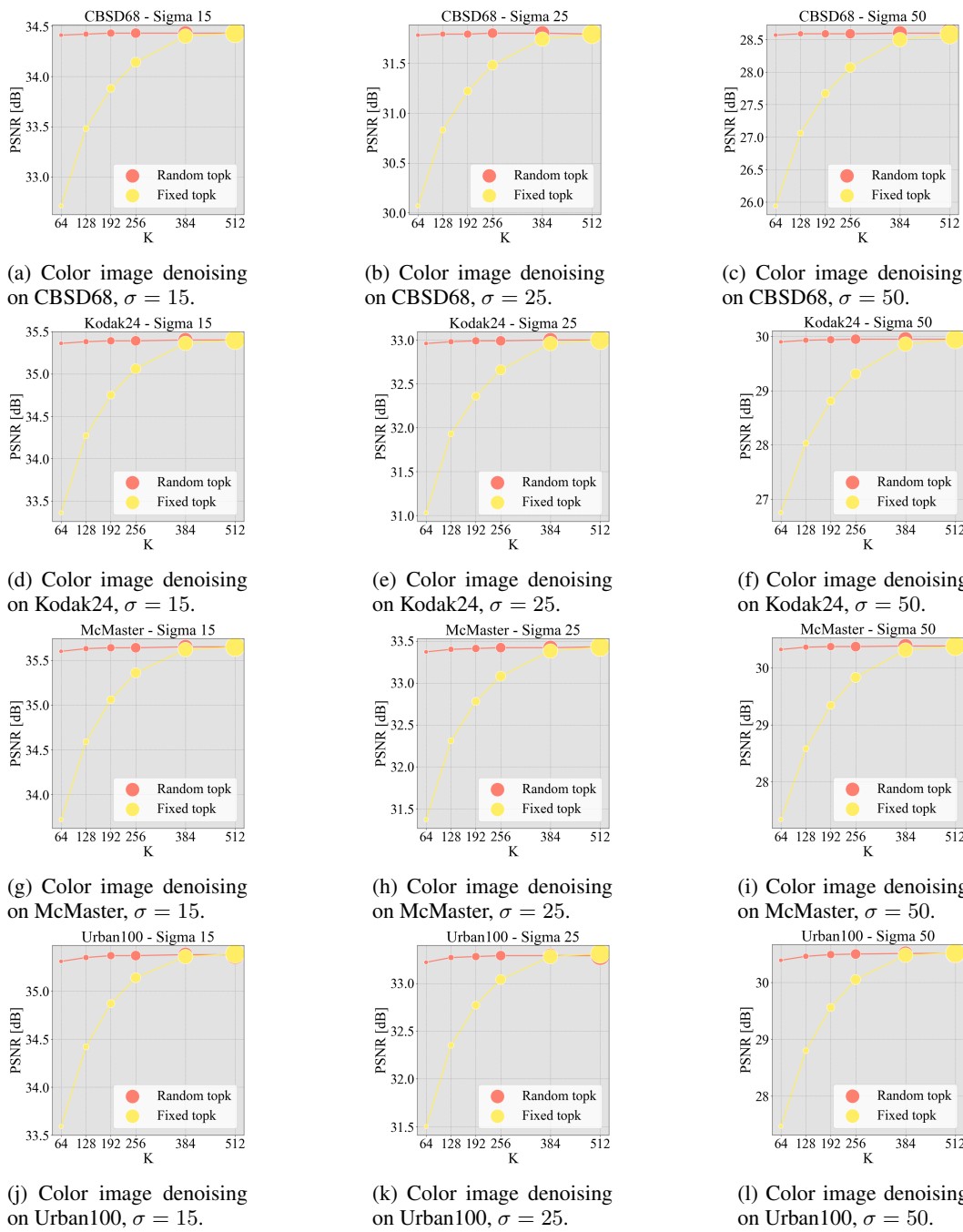

Figure 5: Ablation study on the impact of $K$ for **Color Image Denoising** on CBSD68, Kodak24, McMaster, and Urban100 datasets with $\sigma = \{15, 25, 50\}$.

counterpart. In addition, a decent result can be obtained for the fixed topK setting only when the K is set to the same (*i.e.,* 512) during the inference.

The results of the image SR in terms of hyper-parameters K under different training settings during inference for color images with different scale factors (*i.e.,* x2, x3, and x4) are also provided in Fig. 7. All experiments are conducted in various datasets (*i.e.,* BSD100, Manga109, Set5, Set14, and Urban100). It shows that for datasets like BSD100, Set14, and Urban100, a similar trend can be also observed in Fig. 7 compared to JPEG CAR and image denoising tasks, *i.e.,* the random topK setting performs more stable regardless the change of the K during the inference. However,

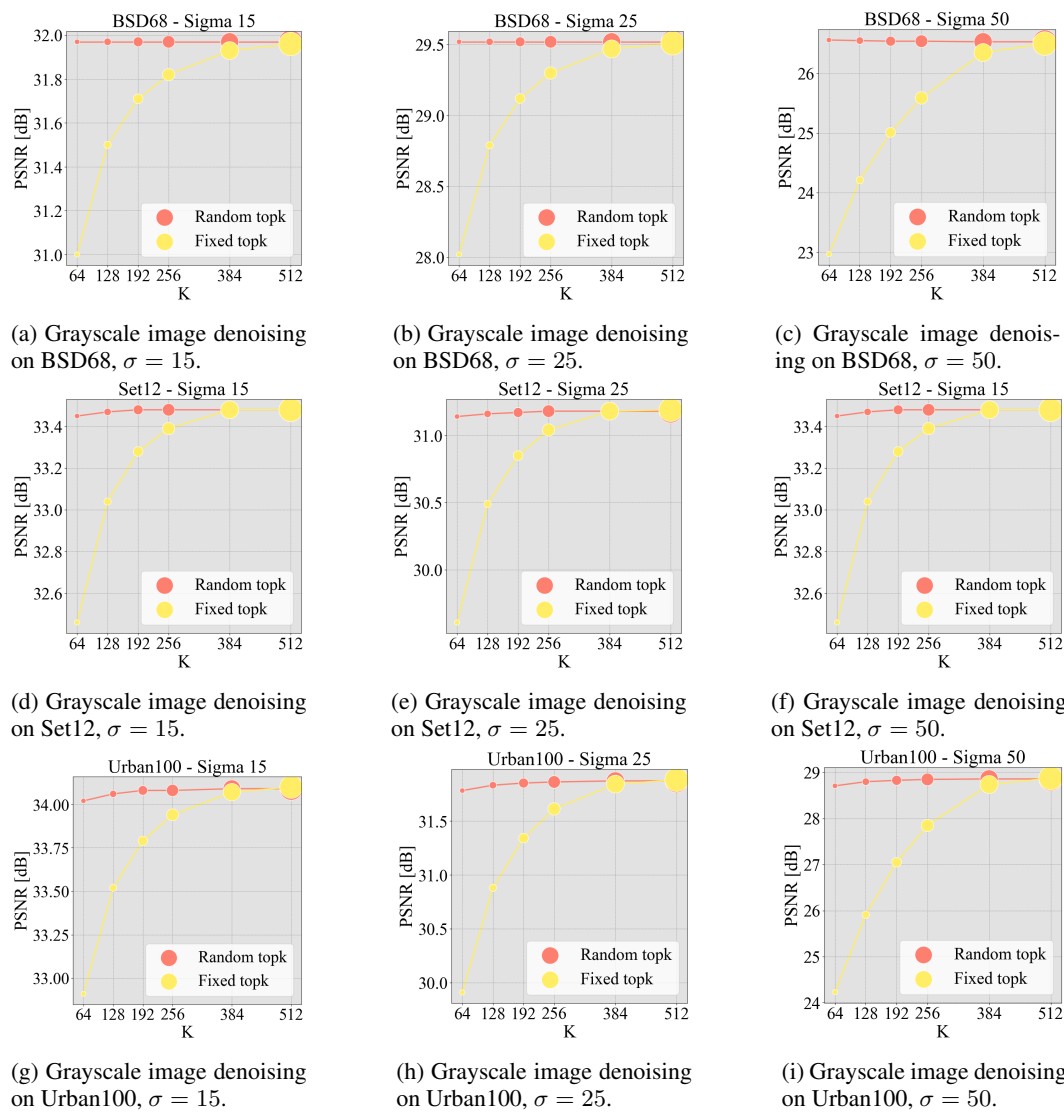

Figure 6: Ablation study on the impact of $K$ for **Grayscale Image Denoising** on BSD68, Set12, and Urban100 datasets with $\sigma = \{15, 25, 50\}$.

for Manga109 and Set5 dataset. The best PSNR is obtained by the fixed topK setting (in (d) - (i) in Fig. 7).

In general, based on all the experimental results mentioned above, we conclude that (1) the random topK setting performs better than the fixed K setting, and usually outperforms the latter by a large margin when K is fixed to small values (*i.e.*, 64, 128, 192, or 256.). (2) For the fixed topK setting, if K is set to big enough (*i.e.,* 512) during inference, the fixed topK setting can also achieve comparative performance or even better performance compared to the random topK setting for several experiments (*e.g.,* color JPEG CAR in Fig 3 (g) and Fig 3 (k)). However, it is not always possible that the large fixed K setting can be generalized to limited computation resources, and the model trained with large fixed K usually needs the same K for inference to maintain the performance, which leads to heavy computation resources needed even for inference.

To this end, we propose to decouple the way to use K between training and inference. *i.e.,* we can use the random sample K during training while an optional fixed K during inference without degenerating the overall performance. It makes it possible to deploy models that heavily rely on large GPU memory during training but to limited GPU resources while maintaining reliable performance during inference. This is also consistent with the way we implement the proposed K-Graph attention

block (*i.e.,* we adopt a *Torch-MasK* version that requires affordable large GPU memories during training compared to *Torch-Gather* while adopting the *Triton* version during inference.)

**Convergence Visualization.** The training log of the proposed KGT for image SR is shown in Fig. 8. The log is reported for the PSNR on the Set5 dataset during training. Two versions of the proposed method including KGT-S and KGT-B are shown in this figure. As shown in this figure, the proposed network converges gradually during the training.

# 4 MORE VISUAL RESULTS

To further support the effectiveness of the proposed KGT intuitively. We provide more visual comparison in terms of JPEG CAR, image denoising, image SR, and image deblurring below.

## 4.1 JPEG COMPRESSION ARTIFACT REMOVAL

For JPEG compression artifact removal, the visual results for grayscale images on the Urban100 dataset are shown in Fig. 10 and Fig. 11, respectively. The proposed method achieves state-of-the-art performance in removing the blocking artifacts in the input images.

## 4.2 IMAGE DENOISING

The qualitative results for image denoising on the BSD68 and the Urban100 dataset are shown in Fig. 12 (grayscale image) and Fig. 13 (color image). It's clear that for both the grayscale and color inputs, the proposed KGT can remove the noise in the noisy input images and recover more realistic textural details in the restored images.

## 4.3 IR ADVERSE WEATHER CONDITIONS

The qualitative results for IR in AWC on the Test1 Li et al. (2020; 2019) dataset are shown in Fig. 9. It shows a challenging case but our method can restore better structural content and clearer details.

## 4.4 IMAGE SR

The comparison of visual results of different image SR methods is shown in Fig. 14 and Fig. 15. Fig. 14 shows the results on the Urban100 dataset, and Fig. 15 shows the results on the Manga109 dataset. The proposed KGT can restore more missing details in the LR images compared to other state-of-the-art methods like SwinIR, ART, CAT, and EDT.

## 4.5 IMAGE DEBLURRING

The visual results for single image motion deblurring are shown in Fig. 16. As shown in this figure, the proposed method can effectively remove the motion blur in the input images and restore more details such as the facial contour, and the characters compared to MPRNet or Restormer.

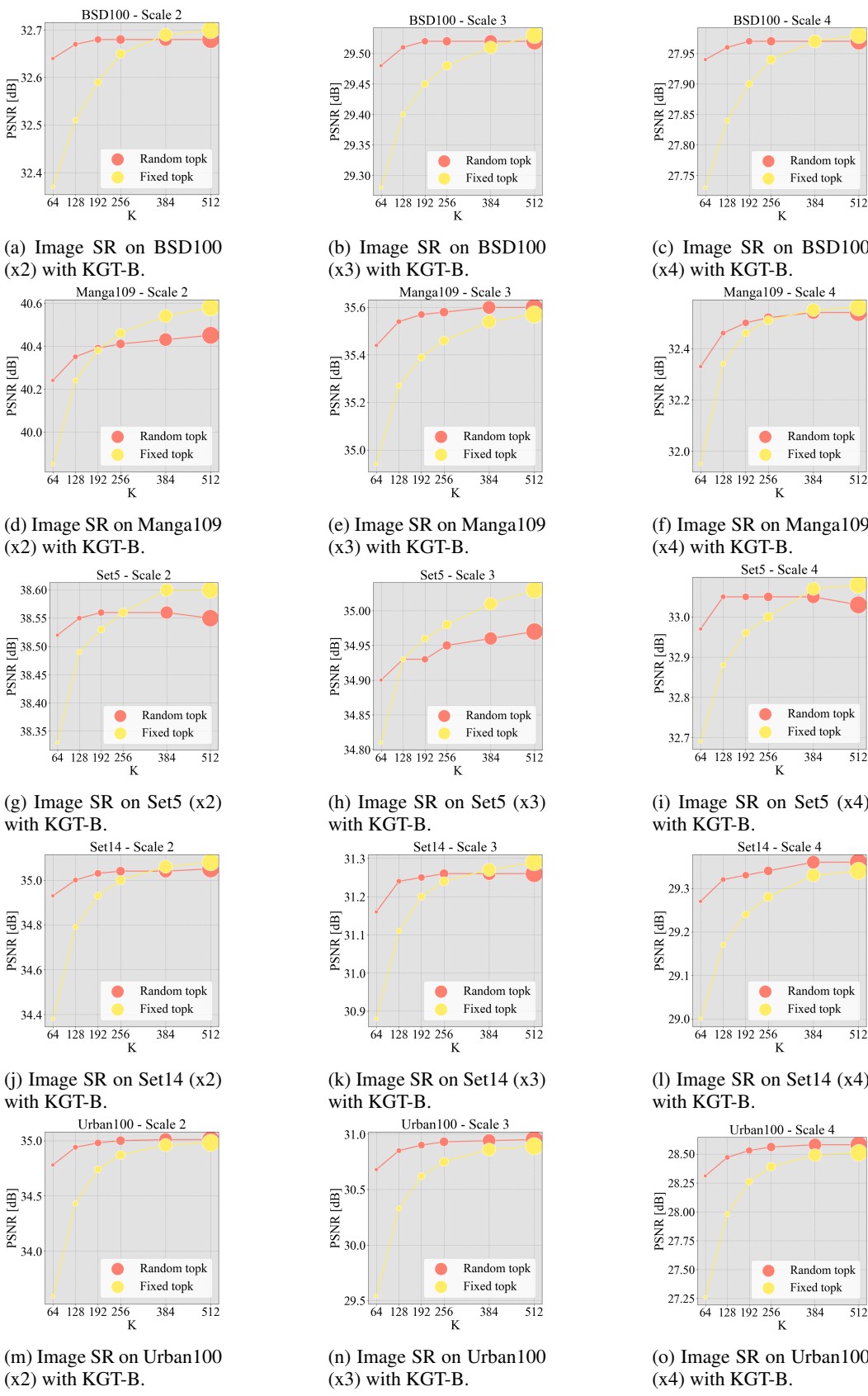

Figure 7: Ablation study on the impact of $K$ for **Image SR** with **KGT-B** on BSD100, Manga109, Set5, and Urban100 datasets with $scale = \{x2, x3, x4\}$.

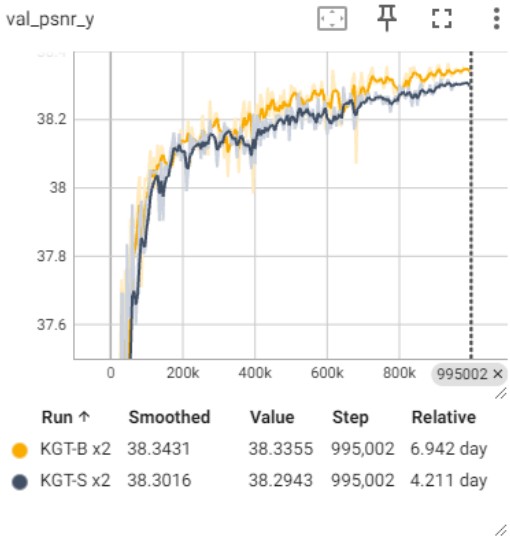

Figure 8: Training log shows the convergence of the proposed KGT during training. Upscaling factor is ×2.

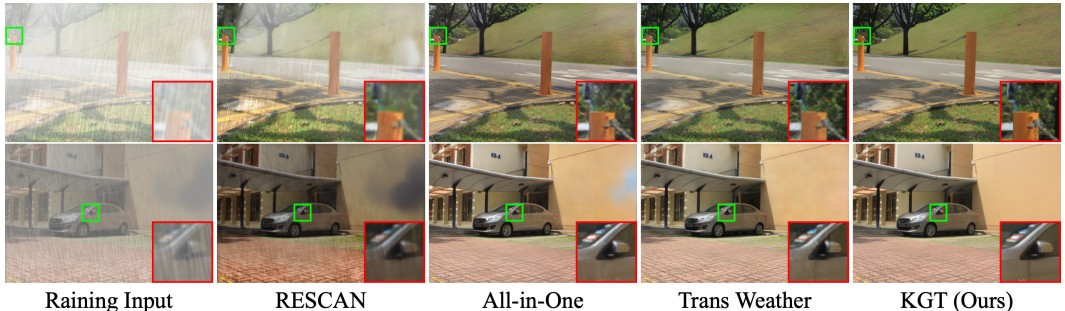

| Raining Input | RESCAN | All-in-One | Trans Weather | KGT (Ours) |

Figure 9: Visual comparison for restoring images in AWC. Best viewed by zooming.

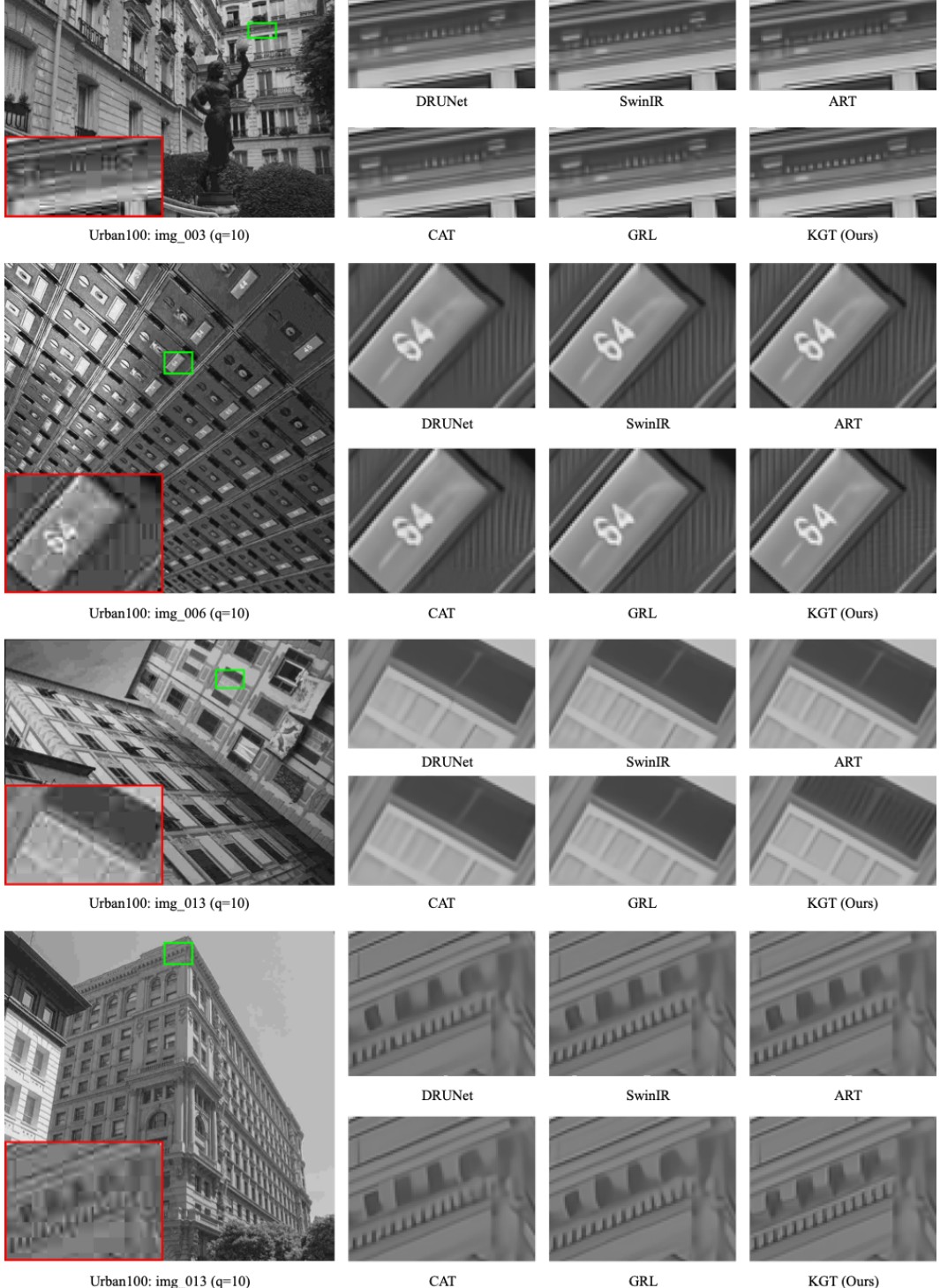

Figure 10: Visual comparison of grayscale JPEG CAR on Urban100 dataset. Best viewed by zooming.

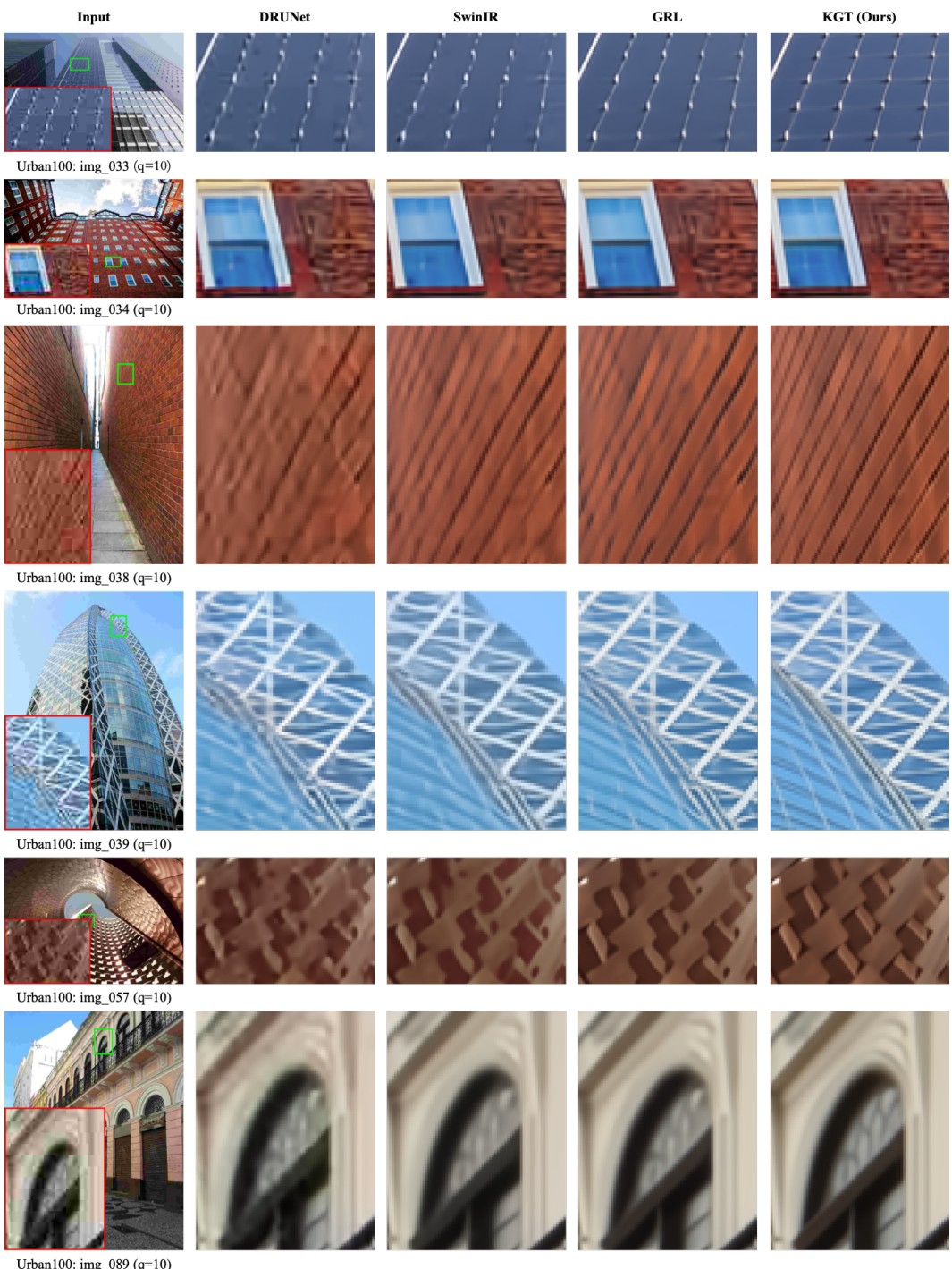

Figure 11: Visual comparison of color JPEG CAR on Urban100 dataset. Best viewed by zooming.

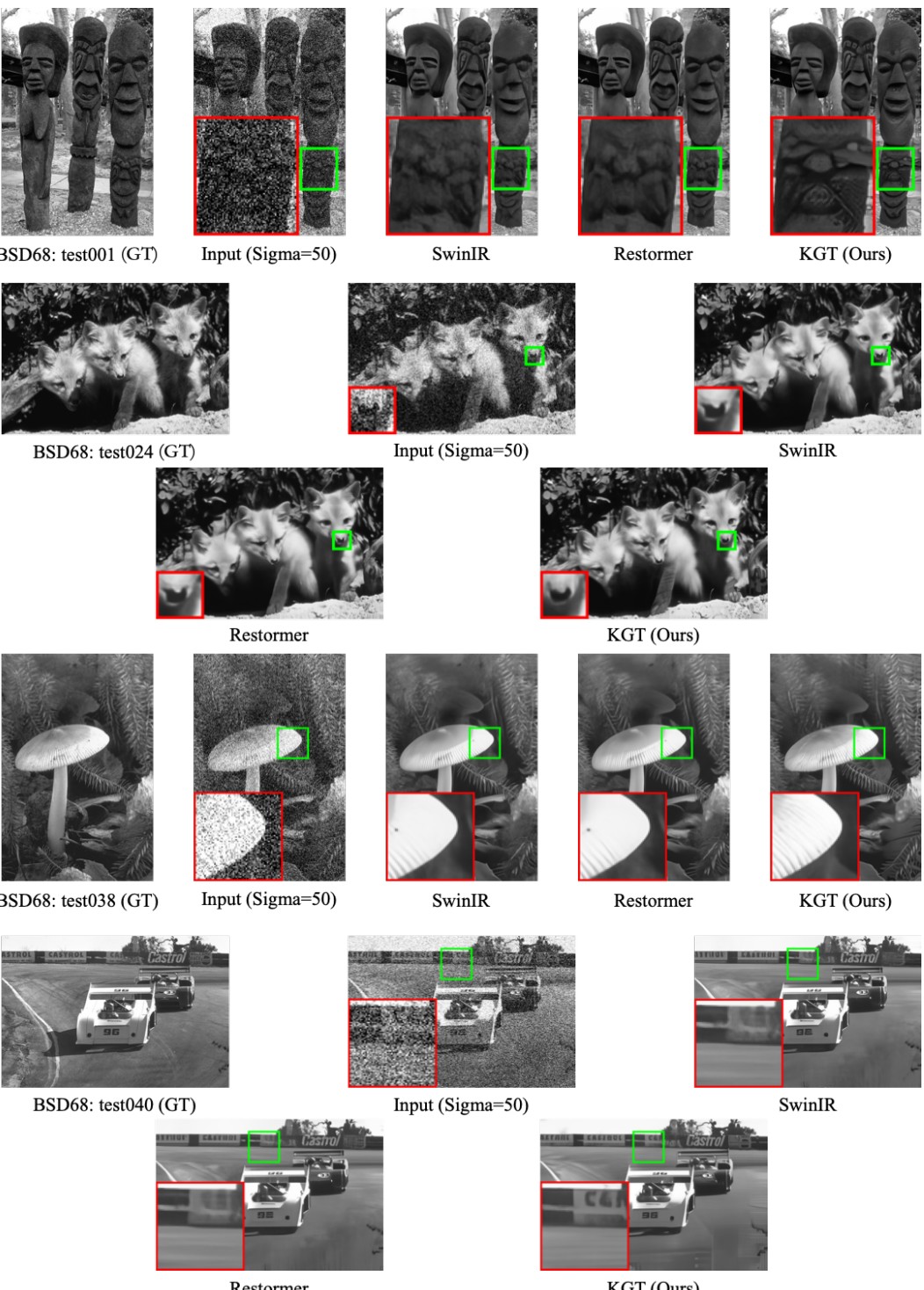

Figure 12: Visual comparison with image denoising on BSD68 dataset. Best viewed by zooming.

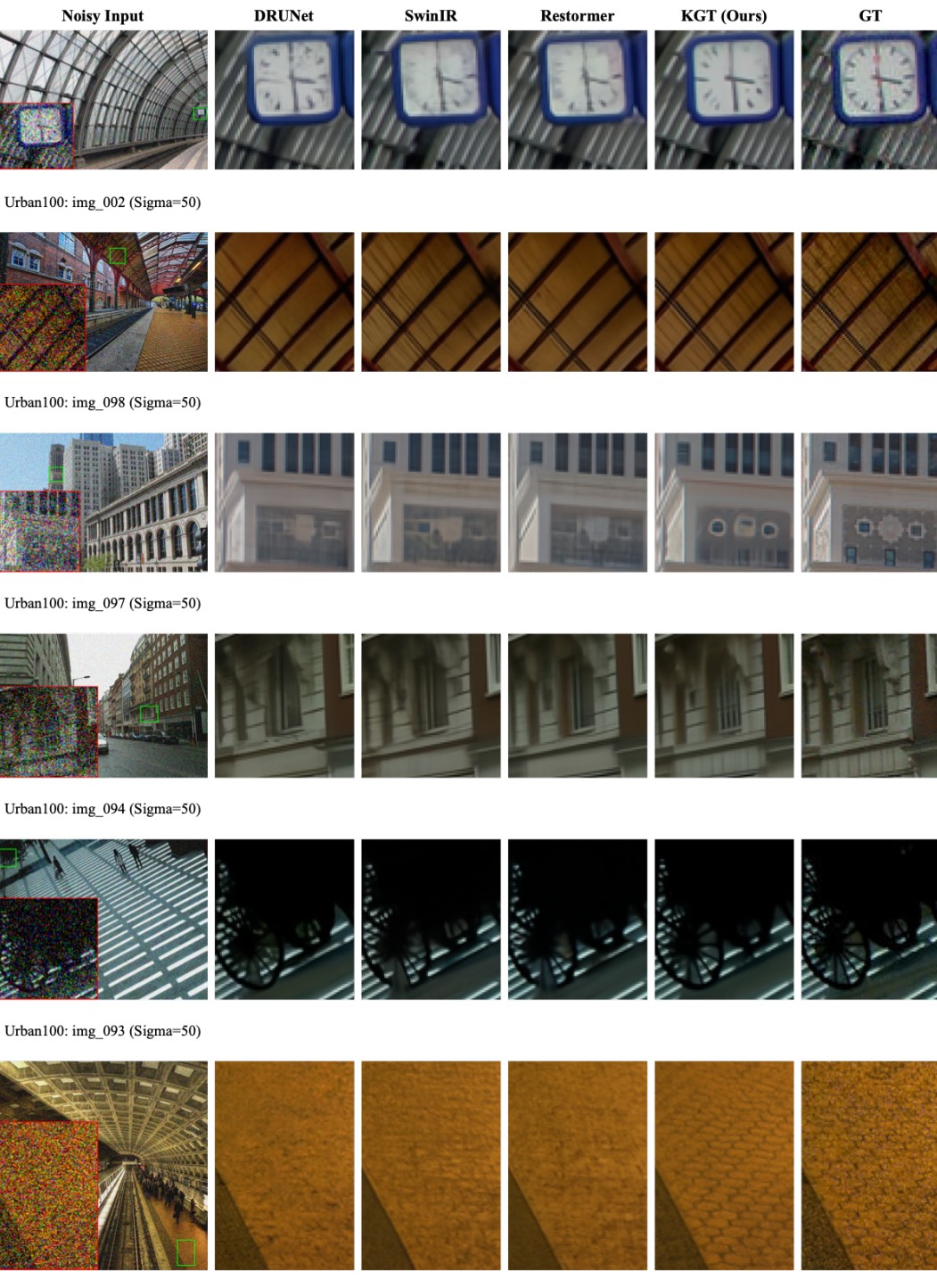

Figure 13: Visual comparison with image denoising on Urban100 dataset. Best viewed by zooming.

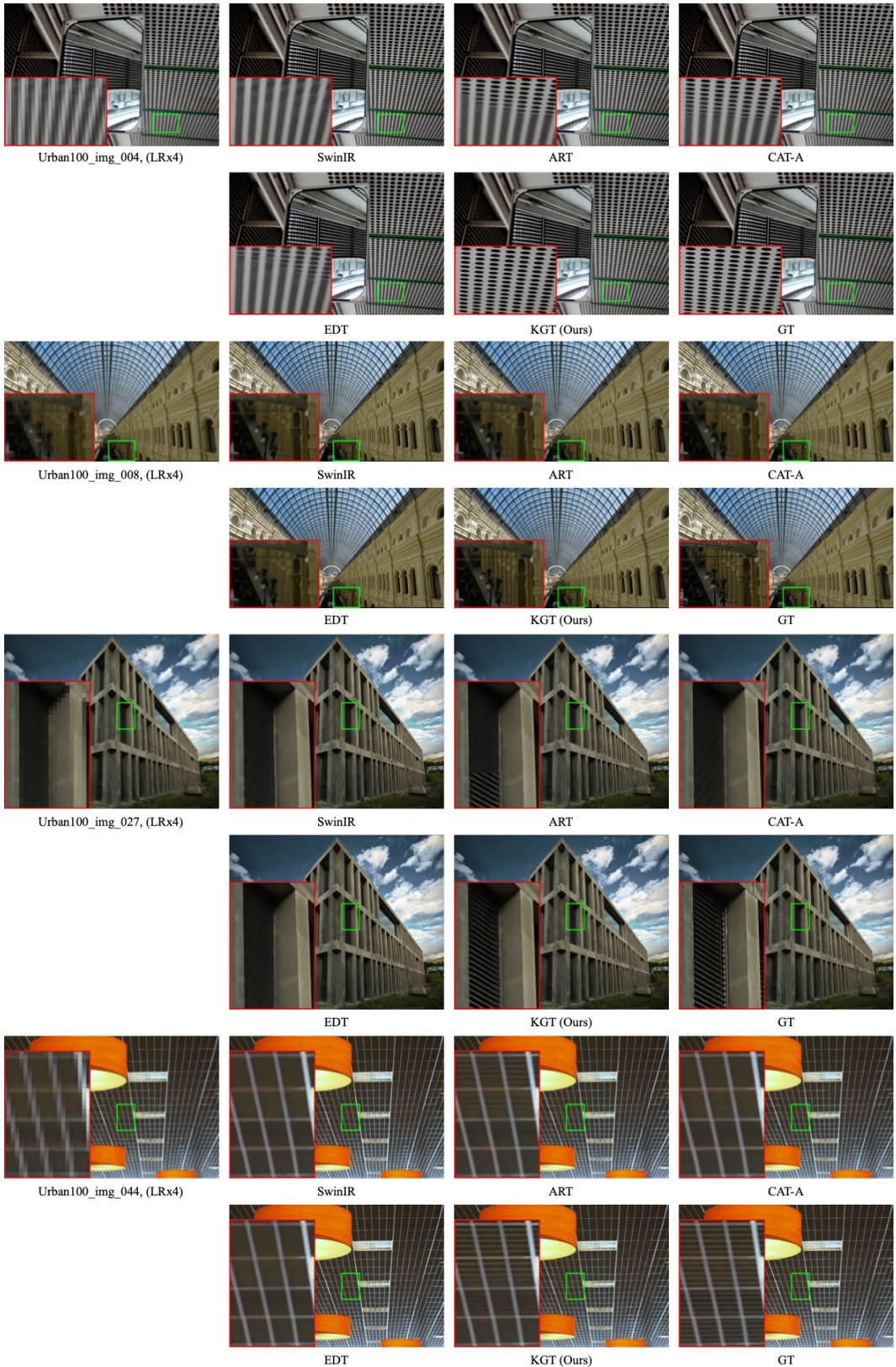

Figure 14: Visual comparison (×4) with image SR on Urban100 dataset. Best viewed by zooming.

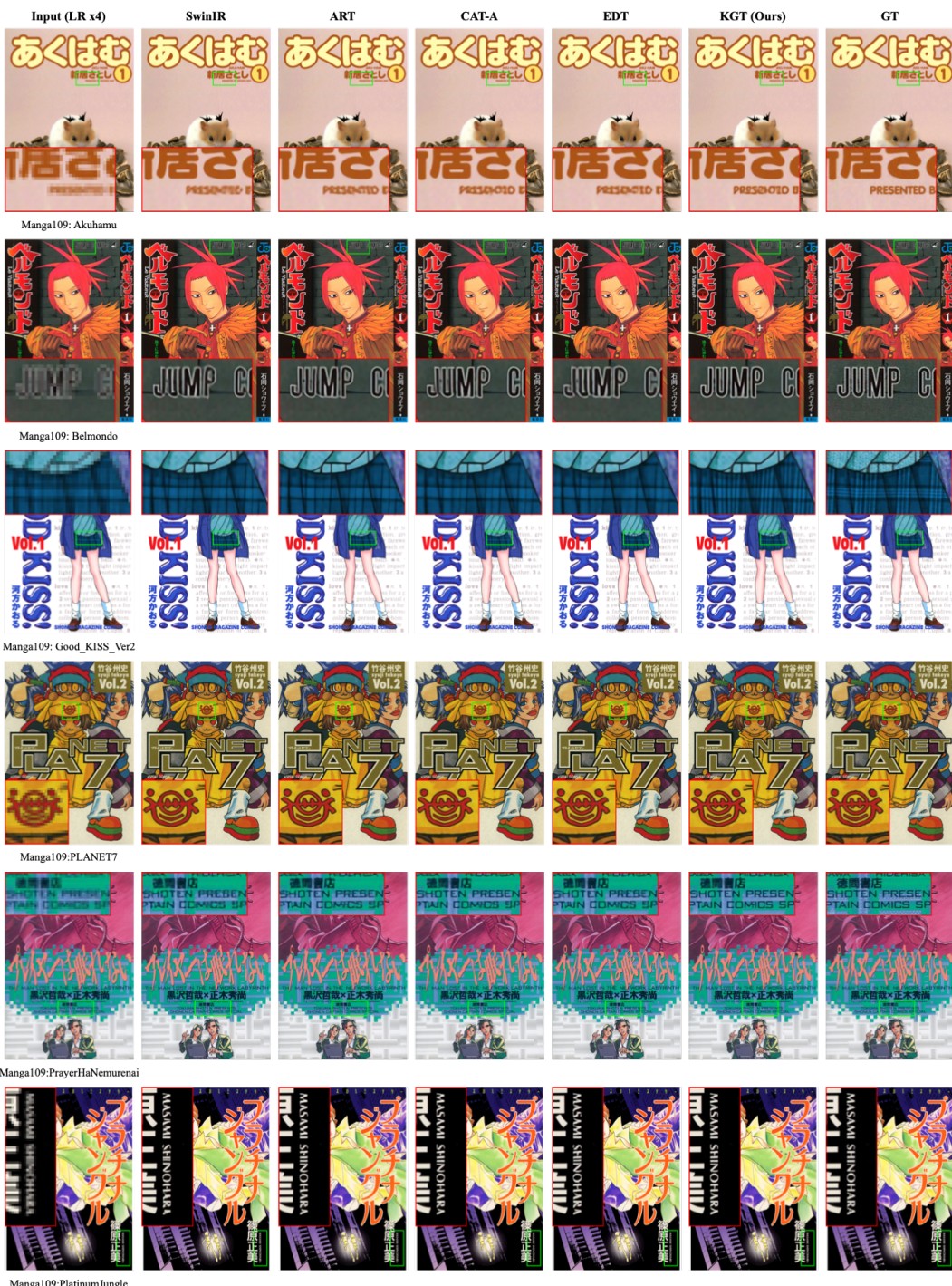

Figure 15: Visual comparison (×4) with image SR on Manga109 dataset. Best viewed by zooming.

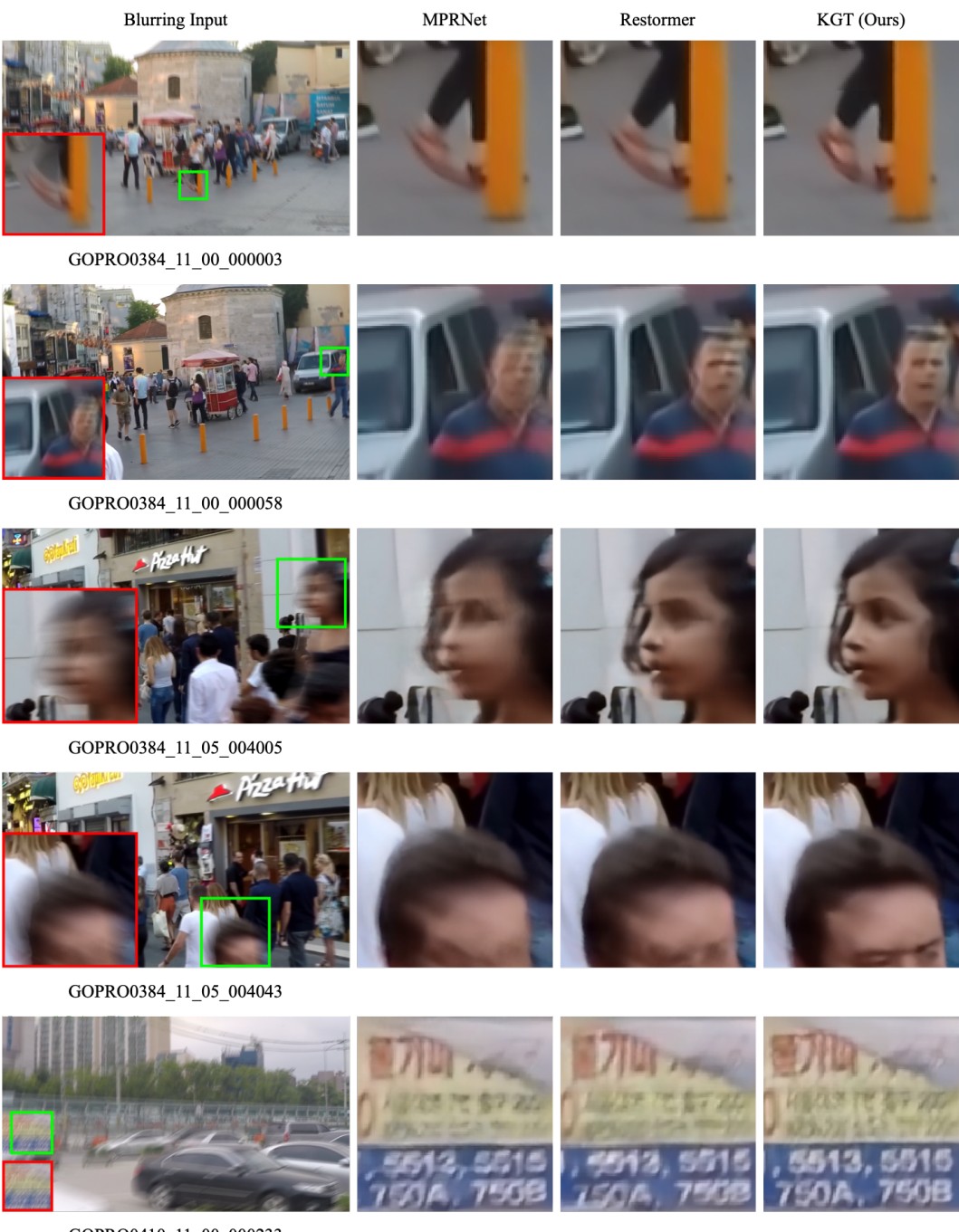

Figure 16: Visual comparison with single image motion deblurring on GoPro dataset. Best viewed by zooming.