# OpenReview forum: "Key-Graph Transformer for Image Restoration"
_ICLR.cc/2024/Conference — Submitted to ICLR 2024_

### Official Review · Reviewer_29bC · 2023-10-31

**Soundness:** 3 good
**Presentation:** 2 fair
**Contribution:** 3 good
**Rating:** 6
**Confidence:** 2

**Summary:**

This proposes a graph based method to capture strong neighborhood cues from images and make them amenable to be fed to a transformer based model. The Q to K projection is used to reduce projection dimension (implicitly) which is learned through a similarity based adjacency matrix formulation. Overall, this is an interesting idea, but how to learn the neighborhood term 'K' needs more looking into since it can be changed from training to inference time.

**Strengths:**

The idea is good and the way to implement it using existing techniques works. The primary challenge of implementing the Key-Graph-Attention block has been implemented by three different techniques. The experiments do show that the method improves the SOTA performance for Image Restoration and other related tasks.

Originality: I think the idea is original and novel.
Clarity: the idea has been developed clearly and important terms in the process have been defined / clarified as necessary.
Quality: I believe the experiment reporting quality could have been improved.
Significance: The method is novel, and the experiment results are strong. This work can be useful for other applications as well.

**Weaknesses:**

The ablation studies were a bit hard to follow. For the randomly sampled strategy, it is not clear whether the value of K is selected once and then training is performed, or is it sampled per batch / epoch? For inference side, 'K is set to desired value for both sets'. I could not understand what is 'desired value'. For Fig 4, the red and yellow circles are becoming bigger in size from left to right. What does it signify. What is the X axis, is it K at inference time?

For the comparative experiments, it is not clear whether the methods were independently implemented, or their results obtained from the papers. If they are as reported then they should match the papers, and if they are all independently implemented then more discussion should be provided for the parameter selection for these methods.

**Questions:**

Please improve the overall experimentation reporting. Segregate the datasets, methods and their implementations and the tasks and create a more cohesive experimentation reporting. This will definitely help this paper.

---

> ### Author Response · Authors · 2023-11-16
> **Response to Reviewer 29bC**
>
> ### Q1: The ablation studies were a bit hard to follow. For the randomly sampled strategy, it is not clear whether the value of K is selected once and then training is performed, or is it sampled per batch / epoch? For inference side, 'K is set to desired value for both sets'. I could not understand what is 'desired value'.
> **A:** Sorry for causing the confusion. The setup of this ablation study is explained as follows. Two sets of experiments are conducted to study the influence of the hyper-parameter K. In the first set, K was held constant at 512 throughout the training process (fixed topK, yellow color), while in the second set, K was randomly sampled from the values [64, 128, 192, 256, 384, 512] (random topK, red color) per training batch. For both sets, K was configured to the specified values such as 64, 128, 192, 256, 384, and 512 during the inference phase. The random topK setting can maintain a relatively stable performance score under different K during inference (We show the comparison in Fig. 4 of our main manuscript and  Supp. Mat.),  This enables various choices of the K  value during inference without a large performance drop.  We mean this kind of choice desired K value. More discussion and analysis can also be found in Section 3 of our update Supp. Mat.
>
>
> ### Q2: For Fig 4, the red and yellow circles are becoming bigger in size from left to right. What does it signify? What is the X axis, is it K at inference time?
> **A:** Thanks for helping us clarify Fig. 4. We have re-organized the caption of Fig.4. The size of the red/yellow circles denotes the FLOPs (We also addressed this in Fig.4 of our updated main manuscript), which means that the FLOPs increase with the number of the K values in the proposed K-Graph Constructor. The X-axis represents the value of  K during inference.
>
>
> ### Q3: For the comparative experiments, it is not clear whether the methods were independently implemented, or their results obtained from the papers. If they are as reported then they should match the papers, and if they are all independently implemented then more discussion should be provided for the parameter selection for these methods.
> **A:** The results of all the other comparison methods are the ones reported in the corresponding papers.
>
>
> ### Q4: Please improve the overall experimentation reporting. Segregate the datasets, methods and their implementations and the tasks and create a more cohesive experimentation reporting. This will definitely help this paper.
> **A:** Thank you for your valuable suggestion. We have refined the experimentation reporting for enhanced clarity and cohesion. You can find the detailed breakdown of datasets, and methods along with their training details, and specific tasks in our updated Supp. Mat. under the section titled "Experimental Reporting."  In addition, we also added the implementation of the proposed Key-Graph Transformer Layer in Section 3.3 of our updated main manuscript. Discussions about the impact of the implementation of the Key-Graph Attention are provided in Section 4.1 of our updated main manuscripts.

---

### Official Review · Reviewer_1vRJ · 2023-11-01

**Soundness:** 3 good
**Presentation:** 3 good
**Contribution:** 3 good
**Rating:** 6
**Confidence:** 5

**Summary:**

This paper proposes a K-Graph Transformer (KGT) for image restoration.  The K-Graph Attention Block is proposed within each KGT layer to conduct the self-attention operation only among these selected nodes with linear computational complexity.

**Strengths:**

+ The technique of this work is somewhat appealing, because K-Graph Attention has low computing cost and high scalability.

+ The experimental evaluation and discussion are adequate, and the results convincingly support the main claims.

+ The paper is well-organized and clearly written.

**Weaknesses:**

- The author should discuss with relevant methods. Chen et al. [1] proposed a top-k strategy to selectively choose the most relevant tokens for image restoration (e.g., deraining).

Ref[1]: Learning A Sparse Transformer Network for Effective Image Deraining, CVPR2023.

- The limitations of the proposed method should be discussed.

-----------------------After Rebuttal---------------------------

Thank you for your feedback. The rebuttal addressed my concerns well. Considering other reviews, I decide to keep my score.

**Questions:**

See the above Weaknesses part.

---

> ### Author Response · Authors · 2023-11-16
> **Response to Reviewer 1vRJ**
>
> ### Q1: The author should discuss relevant methods. Chen et al. [1] proposed a top-k strategy to selectively choose the most relevant tokens for image restoration (e.g., deraining).
> **A:** Thanks for the suggestion. DRSformer [a] proposed a topK strategy that chooses the most relevant tokens to model the non-local priors for deraining. However, the KNN selection in DRSformer occurs after every standard attention operation, where the self-attention is already calculated in a standard manner with  $\mathcal{O} (hw  \times  hw)$ complexity. On the contrary, the  KNN selection of the proposed method occurs exactly before the self-attention operation of each Transformer layer guided by the constructed K-Graph. This selection ensures there are only K nodes in Key (Key here indicted the Key of Query,  Key Value im the self-attention operation) instead of original hw elements in Key, which makes the complexity of the following self-attention become $\mathcal{O} (hw \times  K)$.  We also added the corresponding discussion in the related work section (Non-local Priors Modeling in IR) of our updated manuscript.
>
> [a] Chen Xiang, et al. "Learning a sparse Transformer Network for Effective Image Deraining". CVPR2023
>
>
> ### Q2: The limitations of the proposed method should be discussed.
> **A:** Thanks for the kind suggestion. This study faces a task-specific limitation, i.e., each image restoration task requires training a separate network. While efforts have been made to train models for varying degradation levels within specific types, such as image denoising or removal of JPEG compression artifacts, this approach still leads to inefficiencies in model training and reduces the utility of the trained networks. A potential future enhancement involves developing a mechanism enabling a network to handle diverse image degradation types and levels. Another challenge is the substantial parameter requirement of the proposed KGT, which operates within a tens-of-millions parameter budget. Deploying such a large image IR network on handheld devices with limited resources is challenging, if not unfeasible. Therefore, a promising research direction is the creation of more efficient versions of KGT, integrating non-local context more effectively, to overcome these limitations. We also added the limitation part of the proposed KGT in the updated Supp. Mat.

---

### Official Review · Reviewer_rJSD · 2023-11-01

**Soundness:** 3 good
**Presentation:** 3 good
**Contribution:** 3 good
**Rating:** 6
**Confidence:** 3

**Summary:**

This paper introduces a novel approach, the Key-Graph Transformer (KGT), which selectively choose top-K nodes, and then the chosen nodes undergo processing by all the successive K-Graph transformer layers. The computational complexity of the method can be significantly reduced from quadratic to linear when compared to conventional attention operations. Extensive experimental results show that the proposed KGT achieves state-of-the-art performance on several tasks.

**Strengths:**

1. This article proposes a novel approach where image features within a specified window are considered as nodes in a graph. For each node, a subset of k remaining nodes is chosen to establish connections, and subsequently, a self-attention operation is performed only among these selected nodes.

2. High algorithm efficiency can be achieved by calculating attention only among the k selected nodes, rather than considering all nodes. This approach significantly reduces the computational complexity from quadratic to linear, in comparison to conventional attention operations.

**Weaknesses:**

1. The paper mentions that compared with traditional self-attention, the time complexity of this method significantly slows down from quadratic to linear. Therefore, it is more prudent to add a chart comparing the convergence speed of various methods to the paper.

2. None of the references in the full text are cited, which makes it very inconvenient to view the documents mentioned in the text.

**Questions:**

1. I am uncertain about the usage of "KK". Upon examining the model diagram, I only observe the notation "K" being used, which raises doubts about the origin or meaning of "KK".

2. Please provide a detailed description of the model architecture, including the number of blocks in each layer.

---

> ### Author Response · Authors · 2023-11-16
> **Response to Reviewer rJSD**
>
> ### Q1: The paper mentions that compared with traditional self-attention, the time complexity of this method significantly slows down from quadratic to linear. Therefore, it is more prudent to add a chart comparing the convergence speed of various methods to the paper.
> **A:** We provide the comparison of different neural network models regarding the parameters, runtime, and performance (measured in PSN) on the Urban100 datasets for image SR in Table 2 (Please refer to Response to Reviewer ruU5 Part2: Q3). We reported the small version (KGT-S) of our method. It shows that 1) Among the methods, HAT [d], and KGT-S achieve first-class PSNR performances, reaching 28.37dB and 28.34dB, while KGT-S is much faster and has 41.7% fewer parameters than HAT. 2) SwinIR [a] runs a bit faster than KGT-S but with a PSNR loss of 0.89 dB. Compared with ART, HAT, and HAT-S, the proposed KGT-S is faster and more accurate.
>
> Additionally, reporting the convergence speed of other comparison methods requires retraining all these methods from scratch. This is not easy and not feasible to do so.  For the comparison, the quantitative results of these methods are directly adopted from their papers, and the visual results of these methods are from their provided checkpoints or directly the provided visual results. As a remedy, we report the training log of the proposed  KGT with two versions (i.e., the small version  KGT-S and the base version KGT-B). It shows that the proposed network converges gradually during the training. We also include this in our Supp. Mat. (Section 3: More Abation Analyses -> Convergence Visualization, Figure 8).
>
> [a] Liang Jingyuyn, et al., "Swinir: Image restoration using swin transformer". ICCVWorkshop2021
>
> [b] Zhang Jiale, et al., "Accurate image restoration with attention retractable transformer". ICLR2023
>
> [c] Chen Zheng, et al., "Cross Aggregation Transformer for Image Restoration". NeurIPS2022
>
> [d] Chen Xiangyu, et al., "Activating More Pixels in Image Super-Resolution Transformer". CVPR2023
>
>
> ### Q2: None of the references in the full text are cited, which makes it very inconvenient to view the documents mentioned in the text.
> **A:** We appreciate your suggestion and apologize for any inconvenience caused. We have addressed the citation issue in the updated main manuscript. Specifically, for all comparison methods in the experiments section, we provide the corresponding citation for each method when it is mentioned for the first time.
>
> ### Q3: I am uncertain about the usage of "KK". Upon examining the model diagram, I only observe the notation "K" being used, which raises doubts about the origin or meaning of "KK".
> **A:** We appreciate your diligence in examining our model diagram, and apologize for any confusion caused by the typo "KK". It should be "K". We have rectified this typo in the updated main manuscript for a more accurate representation. Thank you for bringing this to our attention.
>
> ### Q4: Please provide a detailed description of the model architecture, including the number of blocks in each layer.
> **A:** We have two base model architectures, the first one (Archi-V1)  is depicted in Fig.1 of our main manuscript which is for image super-resolution while another U-shaped hierarchical architecture  (Archi-V2) is for other IR tasks i.e., image denoising, single image deblurring, removal of JPEG compression artifacts, image demosaicking, and image restoration in adverse weather conditions. The detailed structure of the U-shaped architecture is also shown in our updated Supp. Mat. The detailed number of the KGT stages and the Transformer layers within each KGT stage for both architectures are listed in the following table (Table 3).
>
>
> Table 3. The details of the KGT stages and KGT layers per stage for both architectures.
> |     | Archi-V1 |      | Archi-V2       |        |                |
> |----------------------|----------------|--------|----------------|--------|----------------|
> |                      | KGT-small      | KGT-base | Down Stages   | Up Stages | Final Stage  |
> | Num. of KGT Stages   | 6              | 8        | 4            | 4        | 1            |
> | Num. of KGT layer/stage | 6            | 8        | 6            | 6        | 6            |

---

### Official Review · Reviewer_ruU5 · 2023-11-06

**Soundness:** 3 good
**Presentation:** 3 good
**Contribution:** 3 good
**Rating:** 5
**Confidence:** 2

**Summary:**

The paper introduces the K-Graph Transformer (KGT) for image restoration. The proposed KGT addresses the computational challenges of transformer-based methods by creating a sparse K-Graph that connects only essential nodes, improving efficiency.

**Strengths:**

A novel Key-Graph Transformer was proposed to conduct the self-attention operation only among these selected nodes with linear computational complexity for image restoration.
Extensive experiments were conducted to verify the effectiveness of the proposed KGT.

**Weaknesses:**

On the image deblurring task, the authors should provide more results of comparison with current state-of-the-art works.
The authors should explain why the KGT, which is built upon the structure of SwinIR, possesses more than double the model parameters of SwinIR (25.82M vs. 11.75M). Are the proposed K-Graph Constructor and K-Graph Attention Block really computationally efficient?
The authors should also provide a comparison between their proposed KGT and some representative state-of-the-art methods in terms of latency.

**Questions:**

It is suggested that that the authors consider discussing the differences and similarities with KiT[1] in the literature review part.
In the demosaicking and denoising tasks, it is recommended to include the appropriate citation information in the corresponding table or context. In addition, in the denoising task, the number of model parameters for Xformer is 25.23M, as stated in the corresponding paper.
In the supplementary material, Is there a misquotation in subsection 3.1? Fig.?? should refer to Fig. 7? Furthermore, the authors mentioned that the training contains two phases in single-image motion deblurring task, which seems to be an uncommon training setup.
Reference:
[1] Lee, Hunsang, et al. "KNN local attention for image restoration." Proceedings of the IEEE/CVF Conference on Computer Vision and Pattern Recognition. 2022.

---

> ### Author Response · Authors · 2023-11-16
> **Response to Reviewer ruU5 Part1**
>
> ### Q1: On the image deblurring task, More results of comparison with current SOTA work
> **A:** For the single-image motion deblurring task, we added three early works with three promising recent SOTA methods [a,b,c] in Table 4 of our updated manuscript. Besides, we also provide more visual comparisons in Fig.13 of our updated Supp. Mat., which further validates the performance of the proposed  KGT initiative.
>
> [a] Lee Hunsang, et al. "KNN local attention for image restoration". CVPR2022.
>
> [b] Zhao Haiyu, et al. "Comprehensive and Delicate: An Efficient Transformer for Image Restoration". CVPR2023
>
> [c] Ren Mengwei, et al. "Multiscale Structure Guided Diffusion for Image Deblurring".  ICCV2023
>
>
> ### Q2: Explain why the KGT, which is built upon the structure of SwinIR, processes more than double the model parameters of SwinIR (25.82 vs. 11.75 M). Are the proposed K-Graph Constructor and K-Graph Attention Block really computationally efficient?
> **A:** Thank you for raising this question regarding the significant difference in the number of parameters between KGT and SwinIR. In short, this difference is because SwinIR has a columnar structure without changing of feature map resolution while KGT uses a U-shaped architecture for image denoising like Restormer [a], KiT [b], and DRUnet [c]. The number of channels of U-shaped architecture doubles each time the feature map resolution is reduced by half. Thus, the U-shaped architecture generally has more parameters but lower computational complexity. This leads to an increased number of parameters in KGT (25.82 M) compared to SwinIR (11.75 M).
>
> We use the columnar architecture (without changing the feature map resolution and number of channels) for image SR and the U-shaped architecture for the other tasks including image denoising, image deblurring, and other tasks. The strategy of using multiple architectures is also explored by the previous method [d,e]. Specifically, Fig. 1 of our main manuscript illustrates the base structure tailored specifically for the super-resolution task. The U-shaped hierarchical structure is detailed in Fig.1 of our updated Supp. Mat. We hope these clarifications address your concerns, and we are committed to providing any further information necessary for a comprehensive understanding of our methodology.
>
> We also conducted an evaluation of the floating-point operations (FLOPs) for both SwinIR [f] and our proposed KGT, as presented in Table 1. The analysis reveals that despite SwinIR having fewer training parameters, its FLOPs significantly surpass those of our KGT. This observation serves to emphasize the enhanced computational efficiency of the proposed KGT over SwinIR.
>
> Table 1. The Floating-Point Operations (FLOPs) Comparison Between SwinIR[f] and our KGT.
> |              |    Input Size    | Params |      FLOPs     |
> |:------------:|:----------------:|:------:|:--------------:|
> | SwinIR [f] | [1, 3, 256, 256] | 11.75M | 752.13 billion |
> |  KGT (Ours)  | [1, 3, 256, 256] | 25.82M | 134.57 billion |
>
> [a] Zamir Syed Waqas, et al.  "Restormer: Efficient Transformer for High-Resolution  Image Restoration". CVPR2022
>
> [b] Lee Hunsang, et al. "KNN local attention for image restoration". CVPR2022
>
> [c] Zhang Kai, et al. "Plug-and-play image restoration with deep denoiser prior". TPAMI2021
>
> [d] Chen Liangyu, et al.  "Simple baselines for image restoration", ECCV2022
>
> [e] Li Yawei, et al., "Efficient and explicit modeling of image hierarchies for image restoration". CVPR2023
>
> [f] Liang Jingyun, et al., "Swinir: Image restoration using swin transformer". ICCVWorkshop2021

---

> ### Author Response · Authors · 2023-11-16
> **Response to Reviewer ruU5. Part2**
>
> ### Q3: The authors should also provide a comparison between their proposed KGT and some representative state-of-the-art methods in terms of latency.
> **A:** We compare our method with other 4 recent promising methods SwinIR [a], ART [b], CAT [c], and HAT-S [d] and HAT [d], in the following table for the x4 Super-Resoclusion task on the Urban100 dataset. In order to show the efficiency of the proposed KGT, we report the trainable parameters, the runtime, and the PSNR performance. Note that here we reported the small version (KGT-S) of our method. The following table (Table 2) shows that 1) Among the methods, HAT [d], and KGT-S achieve first-class PSNR performances, reaching 28.37dB and 28.34dB, while KGT-S is much faster and has 41.7% fewer parameters than HAT. 2) SwinIR [a] runs a bit faster than KGT-S but with a PSNR loss of 0.89 dB. Compared with ART, HAT, and HAT-S, the proposed KGT-S is faster and more accurate.
>
> Table 2. The comparison results of different methods regarding the parameters, runtime, and $\times4$ on Urban100 dataseqt for SR.
> | Method | Params (M) | Runtime (ms) | PSNR |
> | -------- | -------- | -------- | -------- |
> | SwinIR [a] | 11.90  | 152.24   |  27.45   |
> |KGT-S (Ours)|12.02|211.42|28.34|
> |ART [b]|16.55|248.26|27.77|
> |CAT [c]|16.60|357.97|27.89|
> |HAT-S [d]|9.62|306.30|27.87|
> |HAT [d]|20.62|368.61|28.37|
>
> [a] Liang Jingyun, et al., "Swinir: Image restoration using swin transformer". ICCVWorkshop2021
>
> [b] Zhang Jiale, et al., "Accurate image restoration with attention retractable transformer". ICLR2023
>
> [c] Chen Zheng, et al., "Cross Aggregation Transformer for Image Restoration". NeurIPS2022
>
> [d] Chen Xiangyu, et al., "Activating More Pixels in Image Super-Resolution Transformer". CVPR2023
>
> ### Q4: Discuss the differences and similarities with KiT [a] in the literature review part.
> **A:** KiT [a] proposed to increase the non-local connectivity between patches of different positions also via a KNN matching between the base patch and other patches but in every local attention operation. This is different from the proposed KGT that only constructs a Key-Graph via a KNN matching among all the nodes at the beginning of each stage. The Key-Graph is shared for all the attention operations of the Key-Graph Transformer Layers within the same stage. This is also discussed in detail in the related work section (Non-local Priors Modeling in IR) of our updated main manuscript.
> [a] Lee Hunsang, et al., "KNN local attention for image restoration." CVPR2022.
>
> ### Q5: In the demosaicking and denoising tasks, it is recommended to include the appropriate citation information in the corresponding table or context.
> **A:** Thanks for the kind suggestion. For demosaicking, we directly added the citation of each comparison method in Table 7. As for denoising, due to the page limit, we added all the citations in the corresponding text description part of our updated main manuscript.
>
> ### Q6:  In the denoising task, the number of model parameters for Xformer is 25.23M, as stated in the corresponding paper.
> **A:** Thanks for the kind suggestion, we have addressed this issue in the updated main manuscript.
>
> ### Q7: In the supplementary material, Is there a misquotation in subsection 3.1? Fig.?? should refer to Fig. 7?
> **A:** Thanks for pointing out the typo, we have corrected it and gave an overall check of both the main manuscript and our Supp. Mat.
>
> ### Q8: The authors mentioned that the training contains two phases in single-image motion deblurring task, which seems to be an uncommon training setup.
> **A:** Yes, we adopt a two-phase training setup, which is also adopted by the previous methods HAT [a] and GRL [b]. In particular, the first phase is used for pre-training the proposed KGT with a small window size. This improves the training efficiency significantly. During the second phase, we increase the window size on the GoPro dataset.
>
> [a] Chen Xiangyu, et al., "Activating More Pixels in Image Super-Resolution Transformer". CVPR2023
>
> [b] Li Yawei, et al., "Efficient and explicit modeling of image hierarchies for image restoration". CVPR2023

---

> ### Author Response · Authors · 2023-11-22
> **Follow-up on Review for Key-Graph Transformer for Image Restoration (ID713)**
>
> Dear Reviewer ruU5
>
> I hope this message finds you well. I wanted to reach out to ensure that you had received our responses addressing the concerns and queries outlined in your review. We greatly appreciate the time and valuable feedback you provided. In our efforts to improve the manuscript, we have diligently addressed each of the points raised in your review.
>
> We understand the demanding nature of your commitments and respect your time. However, if you could spare a moment to revisit our responses, we would be immensely grateful. Your insights have been invaluable to us in refining the paper and ensuring its quality.
>
> Should you have any further questions or require additional clarifications, please do not hesitate to reach out. We are more than willing to provide any further explanations or information that might assist in your reassessment.
>
> Thank you once again for your thoughtful review. We look forward to any further feedback you might have.
>
> Best regards and many thanks

---

### Author Response · Authors · 2023-11-16
**Shared response to all reviewers**

We sincerely appreciate the diligent efforts of all the reviewers in evaluating our work and offering valuable insights and suggestions. Compared with the previous version, the main differences are summarized below:

- We thoroughly revised both the main manuscript and the supplementary materials (Supp. Mat.) to enhance clarity and coherence.
- We moved the visual comparison results of IR in AWC to the Supp. Mat.
- We provided more visual results in the Supplementary Materials to show a more intuitive understanding of the performance of the method.
- We reorganized the experimental reporting supplemented with an overall Table (Tab.2 in the Supp. Mat), facilitating a clearer presentation of our training details.

Furthermore, we have provided detailed responses to each reviewer's specific questions, addressing their concerns and incorporating suggestions where applicable. We are committed to ensuring that our revised manuscript reflects the rigorous attention paid to the reviewers' comments and enhances the quality and comprehensibility of our work.

---

### Meta-Review · Area_Chair_bXat · 2023-12-17

**Metareview:**

The paper introduces a novel self-attention mechanism for use in image restoration with reduced computational complexity. Reviews remain borderline (positive), even after the rebuttal, with specific issues remaining open or inadequately explained and with no reviewer willing to substantially increase their score. After reading the paper myself and considering the reviews + rebuttal, my impression is that this work does - at the moment - not meet the standard for acceptance at ICLR. I do, however, strongly encourage the authors to revise the manuscript and more clearly point out their contribution and possibly conduct a more thorough experimental assessment.

**Justification For Why Not Higher Score:**

The main reason for the current score (and not a higher one) is that during the discussion and rebuttal period, none of the reviewers seemed to be fully satisfied by the author's answers and clarifications such that the scores remained at their initial values.

**Justification For Why Not Lower Score:**

N/A

---

### Decision · Program_Chairs · 2024-01-16

Reject